

# A robust objective function for calibration of groundwater models in light of deficiencies of model structure and observations

Raphael Schneider[1], Hans Jørgen Henriksen[1], Simon Stisen[1]

[1]Department of Hydrology, Geologic Survey of Denmark and Greenland (GEUS), Copenhagen, Denmark

*Correspondence to*: Raphael Schneider (rs@geus.dk)

**Abstract.** Groundwater models require parameter optimization based on the minimization of objective functions describing, for example, the residual between observed and simulated groundwater head. At larger scales, constraining these models requires large datasets of groundwater head observations, due to the size of the inverse problem. These observations are typically only available from databases comprised of varying quality data from a variety of sources and will be associated with

unknown observational uncertainty. At the same time the model structure, especially the hydrogeological description, will inevitably be a simplification of the complex natural system.

As a result, calibration of groundwater models often results in parameter compensation for model structural deficiency. This problem can be amplified by the application of common squared error-based performance criteria, which are most sensitive to the largest errors. We assume that the residuals that remain large during the optimization process likely do so because of either

model structural error or observation error. Based on this assumption it is desirable to design an objective function that is less sensitive to these large residuals of low probability, and instead favours the majority of observations that can fit the given model structure.

We suggest a Continuous Ranked Probability Score (CRPS) based objective function that limits the influence of large residuals in the optimization process as the metric puts more emphasis on the position of the residual along the cumulative distribution

function than on the magnitude of the residual. The CRPS-based objective function was applied in two regional scale coupled surface-groundwater models and compared to calibrations using conventional sum of absolute and squared errors. The optimization tests illustrated that the novel CRPS-based objective function successfully limited the dominance of large residuals in the optimization process and consistently reduced overall bias. Furthermore, it highlighted areas in the model where the structural model should be revisited.

## 1 Introduction

Numerical hydrological models are often complex physically based models that require substantial parametrization and evaluation against independent observations. In virtually all cases, real processes and structures are simplified even in physically based models. As a result, model parameter values rarely represent a directly observable unit, and typically must be estimated through optimization or indirect inversion (for example Poeter and Hill, 1997; Hill and Tiedemann, 2007). Inversion





is widely used in geophysics, also for mapping of distributed subsurface parameters (a general framework suggested by Giudici et al., 2019). In our case, the calibration process is based on the minimization of the discrepancy between model output and observations, resulting in effective parameter values for a given model setup and observational dataset.

The careful selection of calibration targets or objective functions is an important step in the calibration process as they have a large impact on the resulting optimized model (in the context of hydrological modelling see for example Demirel et al., 2018;

Fowler et al., 2018; Gupta et al., 2009, 2012; Krause et al., 2005). Likewise, the selection of informative observations for model evaluation is a critical part of optimization design (for example Danapour et al., 2019; Hartmann et al., 2017; Pool et al., 2017; Seibert and Vis, 2016; for a more general discussion see Gupta et al., 2008).

These issues are particularly relevant for regional and large-scale groundwater models and when large observational datasets of head elevation are used as calibration targets. At larger scales, outside dedicated research catchments or highly investigated

sites, the hydrogeological model will inevitably be a coarse simplification based on crude interpolation of the underlying geological structures and will typically consist of a limited set of units within which hydraulic properties are assumed to be uniform. The uniformity can be circumvented by highly parameterized approaches such as pilot points (RamaRao et al., 1995), resulting in a huge computational burden and risk of overfitting (Doherty, 2003; Fienen et al., 2009), as also demonstrated for the model system used in this work (Danapour et al., 2019). However, most groundwater models and coupled surface-

subsurface models rely on a unit-based approach where the lack of information and the implicit simplification in hydraulic parameter representation will lead to structural model inadequacies (Enemark et al., 2019).

Such groundwater models require datasets of groundwater head observations to adequately constrain the model parameters. Compared to the uncertainty of simulated heads in groundwater flow models, the measurement uncertainty on groundwater heads itself is usually small (Gelhar, 1986; Sonnenborg et al., 2003). Still, certain observations can be unsuitable for parameter

estimation due to several reasons. This can be due to large elevation variations within a model grid, especially in coarser, large-scale models where horizontal model resolution is in the range of tens to hundreds of metres. Within each such model cell, the model can only represent one elevation and groundwater head, not being able to account for small-scale variations due to finer topography. Moreover, small-scale geological features not described in the hydrogeological model or over-simplification of geological units can result in models that cannot represent some measured groundwater heads. Other model uncertainties

include inaccurate boundary conditions, effects of groundwater pumping not accounted for in the models etc. Then there is observation-related uncertainty: A sufficiently large set of groundwater head observations is rarely obtainable within a given project due to both the required field campaigns and long time periods to be covered. Therefore, head observations are often obtained from databases containing historic records of varying, unknown quality and from various sources. Some of these observations can be affected by nearby pumping, or will contain misinformation on e.g. location, observation time, unit,

reference level or the measurement itself. There are guidelines and methods for quantifying the uncertainty of head observations in relation to aspects of well construction, general slope of groundwater head, temporal representation, or in relation to a given groundwater model, depending on grid scale etc. (Henriksen et al., 2003, 2017; Hill and Tiedeman, 2007, chapter 11). These do, however, not always account for model structural errors and assigning individual uncertainties to each





observation point is difficult, which often leads to model aggregated uncertainty estimates. Disentangling different sources of

uncertainty is a challenge in many hydrological model applications (Renard et al., 2010).

This means that not even measurement uncertainty can be determined for many large-scale applications using large observation datasets of not purely scientific origin. However, an estimate of measurement uncertainty is crucial in weighting different observations in calibrations (Ginn and Cushman, 1990; Hill, 1998).

All these observations can potentially guide the parameter optimization and add value to the model evaluation. However, not

all observations will be equally suitable for estimating optimal parameter values. It is important to recognize that the optimal parameter values are not strictly defined as the values that result in the smallest deviations from observations, but the values that best represent the "true" effective parameter values while minimizing the compensation for misrepresentation of measurements and structural model errors. It is not desirable to minimize the model error if this minimization encompasses unrealistic or untrue parameter values that merely compensate most for structural errors. In other words, tuning of parameter

values should not compensate for model inadequacies (for example Antonetti and Zappa, 2018; Motavita et al., 2019; White et al., 2014).

A solution, that sometimes is employed to deal with observations that seem to be outliers or cannot be explained by the conceptual understanding of the hydrologist, is to exclude them from the parameter optimization process or the general model evaluation (Boldetti et al., 2010; Haaf and Barthel, 2018; Højberg et al., 2015; Keating et al., 2010). Besides not being

scientifically appropriate, unless they can be identified by other criteria than residual size, this would result in a false impression of model accuracy or predictive capability. In addition, exclusion or lowering the weight of observations based on model errors prior to calibration can easily lead to misinterpretation, since it will be unknown if a given residual can be significantly reduced during optimization.

Common performance criteria in groundwater modelling contain a summed error term, usually as squared errors to avoid

cancellation of errors with opposite signs (Chai and Draxler, 2014; Krause et al., 2005; Poeter and Hill, 1997). Such performance criteria are sensitive towards outliers, as large errors dominate the objective function, i.e. it is common in groundwater model calibration that the objective function is dominated by a relatively minor group of observations with large residuals. In cases where large initial residuals owe to inappropriate parameter values in an ideal model structure, these residuals can be minimized without significant trade-off with other residuals. However, in regional scale groundwater models,

structural simplification and inadequacy will always cause trade-offs between residuals at different locations. Therefore, large errors should not necessarily be forcing the parameter optimization in a certain direction, if these large errors owe either to a model structural error or to observational outliers. Still, squared error-based performance metrics (for example the root mean square error (RMSE), sum of squared errors (SSE) or Nash-Sutcliffe efficiency (NSE)) are the most common criteria for the evaluation of hydrological models with observations (Gupta et al., 2009).

Based on the challenges outlined above, this study aims at designing an objective function that limits the impact on parameter identification originating from observations that represent conditions not aligning with the scale, structure or boundary conditions of the groundwater flow model. At the same time, the calibration framework should avoid the necessity for





excluding such observations from the optimization and model evaluation. These observations can be informative with respect to model performance evaluation, and should not be omitted based on arbitrary criteria, but can hamper model parameter

identification.

To achieve this goal, we developed an objective function based on the Continuous Ranked Probability Score (CRPS) (Gneiting et al., 2005) for describing deviations between simulated and observed groundwater head. Commonly, the CRPS is used as an evaluation tool for probabilistic forecasts. Here, we suggest a novel use of it in the context of an objective function for groundwater models with large sets of point observations. We explain the CRPS-based objective function, particularly its

inherent benefits in weighing between large and small errors. We then apply the concept to the calibration of two regional scale coupled groundwater-surface water models in Denmark and compare it to results obtained by using traditional objective functions based on mean squared error (MSE) and mean absolute error (MAE).

## 2 Method

The CRPS is a popular evaluation tool for probabilistic forecasts or model simulations (Gneiting et al., 2005). It can be

expressed as

$$\textbf{CRPS} = \int_{-\infty}^{\infty} (\textbf{\textit{P}}_s(\textbf{\textit{x}}) - \textbf{\textit{P}}_o(\textbf{\textit{x}}))^2 d\textbf{\textit{x}} \qquad (1)$$

where $P_s$ is the empirical cumulative distribution function (ECDF) of the ensemble predictions for variable $x$ at a certain timestep, and $P_o$ the respective ECDF of the observation or truth of variable $x$. Usually, the observation is a discrete value; hence, its distribution is a Heaviside step function that changes from 0 to 1 at the observed value. This means the CRPS can

be interpreted as the area between the ECDF of the forecasts and the respective function of the expected value Fig. 1. Its optimum value is zero. Usually, the CRPS is averaged over a series of timesteps. It is attractive as it combines reliability and sharpness of forecasts in one indicator. Furthermore, for deterministic forecasts, the CRPS simplifies to the MAE.

In this paper, we suggest using the CRPS outside of its initially intended scope, the evaluation of ensembles of forecasts or models: Instead of being applied to the value of a state variable across members of an ensemble forecast, the principle of the

CRPS can also be applied to the value of a state variable across a set of locations (observation points), which is evaluated against the output from an individual model. That means, instead of looking at the ECDF of the ensemble forecasts of a certain variable $x$ (for example the groundwater head at a certain point across an ensemble of models), we consider the ECDF of a certain type of predictions across different points in space (for example the groundwater head at several observation points across space in one individual model). For this purpose, since we have a different expected value or observation at each

observation point, we simply consider residuals (model deviations from the observations) instead of absolute values. That leaves us with an expected or optimal value of 0, and the ECDF of the residuals in every single observation point.

With this background, the CRPS can be used just as other deterministic model performance criteria such as the MAE, MSE, or RMSE.





The particular properties of the CRPS, and its differences to the MSE are illustrated in Fig. 1 (inspired by Hersbach, 2000),
which shows the empirical cumulative distribution of a dataset of five predictions. Commonly, in ensemble forecasting, this
would be predictions of the same variable from five members of a model ensemble. In our case, these are residuals of a
simulated variable (groundwater head) at different observation locations. The verification value or truth, which is commonly
an observation of the variable in question, is marked by the red line. When using residuals, the expected or true value is 0. The
CRPS can be represented by the sum of the areas $dx * dP^2$ in the left panel of Fig. 1. Accordingly, the CRPS for an empirical
cumulative distribution can be written as

$$CRPS = \sum_{i=1}^{n} dx_i * dP_i^2 \qquad (2)$$

Commonly, the MSE is given as

$$MSE = \frac{1}{n}\sum_{i=1}^{n}(x_i - o)^2 \qquad (3)$$

In the right panel of Fig. 1, $x_i - o$ is represented by each $dx_i$, and $dP_i$ has a constant value of $1/n$ for an empirical cumulative
distribution with $n$ values. Accordingly, the MSE can be represented as the sum of areas $dx^2 * dP$ in the right panel of Fig. 1,
and rewritten as

$$MSE = \sum_{i=1}^{n} dx_i^2 * dP_i \qquad (4)$$

Correspondingly, the MAE is the sum of $dx * dP$ in either panel.

Noteworthy are the different contributions of the single predictions to the aggregated CRPS or MSE value (grey areas in Fig.
1). The single prediction with the highest deviation from the truth is -4 and might be considered an outlier. For the CRPS, that
prediction contributes a relatively small amount to the total CRPS (~21%), whereas for the MSE that prediction accounts for
~72% to the total MSE. In a model calibration using the MSE or RMSE as an objective function, most algorithms will
inevitably focus on improving the model fit for that particular point, as it contributes such a large share to the total objective
function. However, considering that such large deviations between model and observations at certain points often owe to
observation uncertainty or issues with the model structures, this could be undesired behaviour.

The CRPS is much less dominated by the largest deviations. As it squares along $dP$ (Eq. 2), instead of $dx$ as the MSE (Eq. 4),
it assigns a relatively lower weight to the residuals at the outer ends of the ECDF. These residuals are also the largest and, in
our context, often those we have a low confidence in being informative to the model parameter estimation. In contrast, the
CRPS favours sharp and reliable distributions of predictions, because of the relatively large values assigned to predictions
close to the truth, as can be seen in Fig. 1. Therefore, we find it relevant to investigate the use of a CRPS-based objective
function for the optimization of large-scale groundwater flow models.



## 3 Model and data

### 3.1 The two study areas

To illustrate the possible advantages of utilizing a CRPS-based objective functions, we will evaluate the effect of different
objective functions on the calibration of two real-world, regional scale coupled distributed groundwater-surface water models.
The two study cases are situated in Denmark: The catchment of the river Storå in western Jutland, and the catchment of the
inner Odense Fjord on the island of Funen, which is dominated by the river Odense Å (Fig. 2). Both catchments are
approximately 1,000 km$^2$ large, with a generally gentle topography ranging from sea level to approximately 120 m.
Geologically, the Storå catchment is more dominated by sandy soils especially in its western part, whereas the Odense
catchment is dominated by clayey soils.

### 3.2 Hydrologic modelling framework

Both models were set up within the MIKE SHE hydrologic modelling software (Abbott et al., 1986). MIKE SHE offers a
transient, fully distributed, physically based description of the terrestrial part of the hydrological cycle. It encompasses 3D
subsurface flow, 2D overland flow, routing of surface water in streams, a description of the unsaturated zone and the coupling
of surface and groundwater.

The two models are based on the National Water Resource Model of Denmark (referred to as DK-model) developed at the
Geological Survey of Denmark and Greenland (Henriksen et al., 2003; Højberg et al., 2013), representing submodels of the
national model. Those two submodels were originally setup within a project exploring possibilities of further developing the
DK-model (Stisen et al., 2018). Like the DK-model, the horizontal grid resolution is 500 m. The description of the subsurface
is based on a hydrogeological model covering all of Denmark. The parameterization is unit-based, i.e. each hydrogeological
unit is assigned homogenous model parameters for hydraulic conductivity etc. For reasons of computational efficiency, some
layers of the hydrogeological model have been combined, resulting in seven computational layers in the Storå model, and ten
in the Odense model. Meteorological forcing is available as gridded datasets with a daily timestep. Further forcing include
groundwater extractions for drinking water supply and irrigation. The models are run with flexible timesteps, allowing a
maximum timestep of 24 hours.

### 3.3 Hydrologic model calibration data

As for the DK-model, the model was evaluated against groundwater head observations from a series of boreholes and runoff
timeseries at stream stations displayed in Fig. 2.

The dataset of groundwater head observations consists of borehole and groundwater level information from the public national
Danish database for boreholes and groundwater, Jupiter (https://eng.geus.dk/products-services-facilities/data-and-
maps/national-well-database-jupiter/). This data has historically been used in the context of the DK-model. Moreover, within
the above-mentioned development project, additional groundwater head observations from boreholes were collected from





regional authorities (Stisen et al., 2018), which are also being used in the context of this paper. In total, the dataset for the Storå model contains groundwater heads from 890 wells (some boreholes have several observations in time and from different depths), offering a total of 5,218 individual head observations. Only 30 of those wells offer more than ten observations in time. For the Odense model, data exists from 1,820 wells, with 44,273 individual head observations. 205 of those wells offer more than ten observations in time.

Model simulations were also evaluated against discharge data, based on timeseries of observed stream discharge with daily resolution from six stations in the Storå catchment, and nine in the Odense catchment.

### 3.4 Hydrologic model calibration setup

The models were inversely calibrated using the PEST software package for parameter estimation (Doherty, 2015). All calibration and evaluation exercises were performed for the period 2000 to 2008, with a warm-up period starting in 1990. The objective function was a weighted aggregation of discharge performance in streams and groundwater heads, and minimized using the Gauss- Marquardt-Levenberg local search optimization method.

Discharge performance was included by using the NSE

$$NSE = 1 - \frac{\sum_{t=1}^{T}\left(Q_{sim}^{t}-Q_{obs}^{t}\right)^2}{\sum_{t=1}^{T}\left(Q_{obs}^{t}-\bar{Q}_{obs}\right)^2} \tag{5}$$

where $Q_{sim}^{t}$ and $Q_{obs}^{t}$ are the simulated and observed discharge at timestep $t$, and $\bar{Q}_{obs}$ the mean observed discharge. NSE values range from $-\infty$ to 1, with 1 indicating a perfect match. Being based on squared residuals, the NSE generally is more sensitive towards peak flows, and less to the overall water balance. Based on this and previous experience with the DK-model (Højberg et al., 2015), also the water balance

$$fbal = \frac{\bar{Q}_{obs}-\bar{Q}_{sim}}{\bar{Q}_{obs}} \tag{6}$$

and the water balance for the low-flow summer period

$$fbal\_s = \frac{\bar{Q}_{obs,s}-\bar{Q}_{sim,s}}{\bar{Q}_{obs,s}} \tag{7}$$

were included. In the latter, $\bar{Q}_{obs,s}$ and $\bar{Q}_{sim,s}$ are the mean observed and simulated flow for the summer months June, July and August.

The groundwater observation dataset consists of observations of groundwater heads from boreholes. Each borehole can contain one or several observations in time, and can contain one or several screen depths, i.e. observations from different geological layers. It was chosen to aggregate all observations within each model grid cell, which is the smallest unit the model can resolve. Weighting then was applied according to the number of observations per grid cell: A relative weight of 1 was applied to cells with one observation, a weight of 2 to cells with two to nine observations, a weight of 3 to cells with ten to 99 observations, and a weight of 5 to cells with 100 or more observations. Then, the model was evaluated based on the mean error of all the observations within each model grid cell across time. Three different objective functions were compared:

First, a sum of squared errors (SSE)





$$SSE = \sum_{i=1}^{n} (ME_i)^2 \tag{8}$$

where $ME_i$ is the mean error of simulated groundwater heads in each of the $n$ model grid cells containing observations. Second,

a sum of absolute errors (SAE)

$$SAE = \sum_{i=1}^{n} |ME_i| \tag{9}$$

(Note that the terms SSE and MSE, and SAE and MAE can be used interchangeably in this context, as $MSE = 1/n * SSE$.

As MSE and MAE are the more common terms, we will use those throughout the rest of the paper.)

And lastly the CRPS-based objective function (see Eq. 1), using the ECDF of all model grid cells' $ME_i$, and a mean error of 0

as expected value.

The different objective function groups were weighted so that the NSE contributed approximately 21%, fbal 8%, fbal_s 4%

and the groundwater heads (CRPS or MSE or MAE) 67% to the aggregated objective function for the initial parameter set. I.e.

the objective function was made up of one third of stream discharge related metrics, and two thirds of groundwater head related

metrics.

In the Storå model eight model parameters were calibrated: six different geological units' hydraulic conductivities, the root

depth, and the saturated zone drain time constant. For the Odense model, the calibration was set up with six free parameters:

four hydraulic conductivities, the root depth, and the drain time constant.

## 4 Results

### 4.1 CRPS in comparison with other common performance criteria

In order to illustrate the implications of using either CRPS, MSE or MAE as the performance criteria for different error

distributions a simple benchmark test was conducted for some example error distributions. Fig. 3 shows those distributions,

and their respective performance criteria values. The expected value (red line) always is 0. The modelled errors' cumulative

distribution function is given by the blue line. For the benchmark (top left), this error distribution is a normal distribution with

a standard deviation with a mean of 0 and a variance of 1. Simply increasing the error (top right) by doubling the standard

deviation, leads to a CRPS and MAE twice as large as in the benchmark, and an MSE four times as large. When a bias is

introduced by changing the mean of the error distribution to 0.5 (bottom left), however, the CRPS shows more sensitive than

the other metrics: Its value increases 1.4-fold over the CRPS in the benchmark, whereas the MSE and MAE only increase 1.2

and 1.1-fold, respectively, over their values in the benchmark. When outliers are being introduced to one end of the distribution

(bottom right) by replacing 10% of the benchmark with absolute value samples from a normal distribution with a standard

deviation of 5, the CRPS shows less sensitive. It only increases 1.2-fold compared to its value in the benchmark. The MSE

increases 3.4-fold, and even the MAE is more sensitive than the CRPS, increasing 1.4-fold compared to its benchmark value.



## 4.2 Results of hydrologic model calibration

Fig. 4 displays the ECDF of the mean errors of groundwater head per grid cell, after calibrations, with results from Storå in
the right panel and Odense in the left panel. Both models were calibrated twice starting from two different initial sets of
parameter values each. Even though the two initial sets of parameter values were deliberately chosen to result in opposite
overall biases in the modelled groundwater heads, the calibrated parameter sets resulted in virtually the same error distributions.
Subsequently, all results shown are based on the calibrations with initial parameter set 1. Clear differences, however, can be
seen dependent on the choice of the objective function: Using the MSE or MAE as an objective function resulted in a larger
bias in the calibrated model than using the CRPS, as also can be seen in Table 1. Not surprisingly, the best value for each
metric is obtained in the calibration where the respective metric is used as the objective function. The lowest ME for
groundwater heads, however, consistently results from the calibration using the CRPS as objective function. This means that
the CRPS-based calibration allows us to find a solution that has a lower bias than the conventional MSE-based or MAE-based
calibration. It does that on the cost of a slightly higher RMSE. However, as discussed before, the value of the RMSE is
dominated by outliers. Hence, we assume that the CRPS calibrations represent an overall better, or more balanced model fit,
as we will discuss in more detail below.

Furthermore, river discharge performance is relatively insensitive towards the use of different objective functions for
groundwater heads – see the lower part of Table 1.

Fig. 5 displays the absolute mean errors of groundwater head in each observed grid cell (y-axis), in relation to the difference
in mean error in each grid cell between the CRPS-based and MSE-based calibration (x-axis). For the majority of all grid cells,
the error after the CRPS-based calibration is slightly lower than after the MSE-based calibration, which places them in the left,
white half of the plot. Only few observations, amongst those many with large errors, exhibit a larger error after the CRPS-
based calibration, which places them in the right, grey half of the plot. This is in line with the results in Table 1 – the CRPS-
based calibration results in a slightly higher RMSE, yet a clearly lower ME. Furthermore, Table 2 shows that the majority of
cells with *small* error (below 5 m) exhibits *smaller* errors after the CRPS calibration than after the MSE calibration – this
applies to 65% of those grid cells in the Odense model, and 62% in the Storå model. In contrast, the majority of cells with
*large* error (above 10 m) exhibits *larger* errors after the CRPS calibration – 88% and 63% of such cells in the Odense and
Storå model, respectively. The same can be shown for a comparison of the CRPS-based with the MAE-based calibration. The
picture only differs marginally from Fig. 5 etc.; hence, the results are not shown here.

The same can be observed in the maps in Fig. 6 and Fig. 7 for the two model domains: The first map of each model shows all
grid cells with observations, colour-coded for their error after the CRPS-based calibration. The second map of each model
displays only those cells which show a lower error in the CRPS-based than in the MSE-based calibration. The third map
displays only those cells which show a larger error in the CRPS-based calibration than in the MSE-based calibration. Large
errors dominate the third map (reddish and blueish colours), while small to moderate errors dominate the second map
(yellowish colours). More interestingly, some distinct areas with almost exclusively large errors become apparent: In the Storå





model, there appears to be specific issues in the very northern end of the model, where we overestimate groundwater heads (blue colours), and in the southwestern corner, where we underestimate groundwater heads (red colours). In the Odense model, there are two distinct regions in the south-east and mid-west, where groundwater heads are underestimated by the model.

## 5 Discussion

The issues arising from squared error-based performance criteria being sensitive to (even few) outliers and extreme residuals are well known (Berthet et al., 2010; Legates and McCabe, 1999; Moriasi et al., 2007; Chai and Draxler, 2014; Chen et al., 2017) and also discussed in time series forecasting in general (Armstrong and Collopy, 1992; Chen et al., 2017). This also led to the recommendation of using multiple performance metrics, as any single metric will provide information only on a certain aspect of the error characteristic (Chai and Draxler, 2014). Also the popular NSE exhibits similar issues of being highly

influenced by outliers, as it is also based on the squared error term (Krause et al., 2005; McCuen et al., 2006). When comparing different evaluations of ensemble streamflow predictions, Bradley et al. (2004), noted that errors, when using MSE-based performance criteria, are dominated by two distinct periods with specific weather patterns. Excluding those two periods leads to significantly different performance criteria value, and different conclusions on the skill of their ensemble predictions. Similarly, Berthet et al., 2010 showed that in flood forecasting, short periods of high flow values with typically large errors

make up the largest part of the value of quadratic metrics of the entire hydrograph. In cases where there are both some large negative as well as some large positive residuals, as is often the case with groundwater head calibration, a squared error-based optimization is virtually limited to finding the best trade-off between these two groups, without much consideration to the majority of residuals which fall within a reasonable range and which should ideally inform parameter identification.

The CRPS-based objective function addresses the limitations of the squared error-based objective functions by putting more
emphasis on the position of each residual along the cumulative distribution than on the magnitude of each residual. Effectively this means that large, less frequently occurring residuals are assigned a relatively lower weight than in squared error-based metrics. It is an underlying assumption that during the optimization process the least probable residuals will coincide with the largest residuals for large datasets.

With some theoretic error distributions (Fig. 3) it can be illustrated that the CRPS is less sensitive to outliers in an error
distribution than squared error or even absolute error-based performance criteria. On the other hand, it is more sensitive to an overall bias of the majority of all errors. If used as an objective function, the CRPS will focus on reaching a balanced result with little bias, while accepting some potential outliers with large residuals. The MSE (and also the MAE), will more likely result in a distribution with some overall bias, if only the largest residuals can be reduced, as those dominate the overall value of the metric. In real world cases, with large observational dataset of mixed and uncertain data quality and complex models
exhibiting inevitable simplifications of processes and structures, we have to accept that a part of the observational dataset will exhibit large residuals. Nonetheless, we still assume that after parameter estimation, the majority of the observations can fall



within a reasonable margin of error, and those large residuals are considered less informative to the parameter estimation process. This is where we can utilize the strength of CRPS-based performance criteria over MSE or MAE-based criteria.

When comparing different performance criteria applied to the parameter estimation in two real-world groundwater-surface
water models, the novel CRPS-based calibrations showed an improvement of the vast majority of residuals – which are mostly small – compared to MSE or MAE-based calibrations, with only few residuals increasing – most of them large (see Fig. 5 and Table 2). With the given model structure based on geological units and units-based parameters, "error hotspots" as visible in Fig. 6 and Fig. 7 are likely to indicate a general issue with the model structure, for example the construction of the hydrogeological model, or boundary conditions. In parameter estimation, we explicitly want to avoid compensating for issues
such as model conceptual errors or observation errors by trying to tweak model parameters. Due to its characteristics of being comparably insensitive to outliers, but more sensitive towards an overall bias, a CRPS-based calibration is less prone to show such undesired behaviour. It will i) maintain and make such error hotspots more pronounced, allowing for easier identification of general issues with model structure, ii) not compensate as much for observations that are inadequate or incompatible with the model structure by tuning parameters values, and iii) obtain best model performance for the vast majority of observations.
This points in the direction of an iterative optimization process where all available observations are included in a parameter estimation, but where observations that represent conditions not consistent with the model structure or have a large error, are not dominating the parameter optimization process. Subsequently, these observations will be evaluated. When they illuminate a systematic model structural deficiency, they can lead to a re-evaluation of the model structure prior to further parameter estimations. In other cases, for example where observations with large residuals are scattered randomly across the model
domain, they can be regarded as not representative of model scale and structure, as observational outliers or simply as indicators of the overall model uncertainty.

In a squared error-based parameter estimation, observations with large residuals dominate the objective function of the estimation. That can lead to locking parameters to unrealistic, non-optimal values. Alternatively, an outlier filtering can be performed. However, with the mentioned large, non-scientific datasets as used in our model case, and when this is done prior
to the parameter estimation, it is hard to determine which observations with large residuals represent observational outliers, or model structural deficiencies, or non-optimal parameter values. Statistically sound outlier filtering often requires timeseries of observations (Jeong et al., 2017; Peterson et al., 2018), whereas our observation dataset is comprised of many observation points with only a single or few observations in time, and only few observation points with a whole time series. Furthermore, outliers can also be detected based on spatial patterns (Bárdossy and Kundzewicz, 1990). However, the dataset's spatial
coverage can be too coarse and scattered compared to the variables actual spatial variability to allow for reliable outlier detection based on spatial patterns or such methods require assumptions about model structure (Helwig et al., 2019) or still rely on a row of subjective criteria (Tremblay et al., 2015). As mentioned in the introduction, it appears that the scientific groundwater modelling literature assumes that data have been through a data control/data validation process where outlies have been identified and removed, before the data are used in a modelling context. However, this is i) not practical in case of
data from thousands of observation wells from different sources with different (and unknown) data quality that are not




originating from scientific monitoring programmes, and ii) risks omitting valuable information about model parameters or model structural deficiencies. For such cases there is a lack of methodology/metrics that can make use of all available data without allowing outliers to dominate.

It is worth highlighting that the CRPS-based objective function proposed here for calibration of groundwater flow models with large observational datasets assumes that, after adequate calibration, remaining large residuals represent areas of either model structural uncertainty or highlight observational errors, and do not dominate the total number of observations. Our real world model applications are examples where this is the case and the tests based on very different initial error distributions (Fig. 4) illustrated that even though starting with large residuals for a large part of the observations (base 2 in Fig. 4) this will change during the parameter optimization.

# 6 Conclusions

In this paper, we suggested the use of a CRPS-based performance criteria in the objective function of hydrological model parameter estimation. The CRPS originates from the evaluation of probabilistic forecasts. To our knowledge, it has not been used before as performance criteria in our context of deterministic model optimization. In addition, there are few examples in the literature of studies that attempt to practically address the well-known issues related to the use of squared error minimization in optimization problems. These problems are specifically relevant for optimizations problems related to uncertain model structures and large uncertain evaluation datasets, both being common in large-scale groundwater modelling.

Many common performance criteria used in model parameter estimation are based on the squared error term (for example RMSE or NSE). When aggregating over a set of observations, squaring errors results in the aggregated value being dominated by the largest errors. Consequently, a parameter estimation minimizing an objective function based on squared errors will focus on reducing the largest errors. Often, the largest residuals are the result of model structural uncertainty or observation uncertainty, i.e. they may not be informative to the parameter estimation process.

The CRPS, however, is less sensitive towards largest residuals than MSE or MAE. Instead, the CRPS is more sensitive to the overall bias of the majority of observations. Such a behaviour is beneficial for many parameter estimation problems. We could show this for two real-world examples of distributed hydrological models, that were calibrated against a set of groundwater head observations. When calibrating the two models against the CRPS of the groundwater head residuals, both models consistently showed lower biases than when calibrated against the MSE or MAE. Along with the lower bias, the CRPS calibration resulted in better model fits at the majority of observation points compared to the MSE or MAE calibrations. When looking exclusively at observations with large errors, the CRPS calibration, in contrast, performed worse than the MSE or MAE calibrations. This, however, is in line with expectations, as we assume that such large error mostly can be explained by model structural errors or observation errors. Hence, such observations are not considered to be informative to the parameter estimations process. Despite this, they still can be informative as they might point out areas of model structural errors, or systematic observation errors.

In practical applications, such as in our example where we used a large dataset of groundwater head observations from a public database with varying and undetermined quality in combination with a regional scale groundwater model, model structural

uncertainty and observation uncertainty are difficult to quantify. The same applies to global scale hydrologic modelling, where often creative calibration targets are used (e.g. Yang et al., 2019). Another example is a calibration of stream flow in the signature domain instead of the time domain, assuming this make the calibration more robust to objective function or data deficiencies (Fenicia et al., 2018; Westerberg and McMillan, 2015).

The work presented in this paper is a practical way of dealing with model inadequacy (or observation uncertainty, or model

structural uncertainty) in cases where those uncertainties are hard to quantify. However, mathematically sound accounting for such effects requires at least some idea of the magnitude of observation uncertainty or model inadequacy, see for example the discussion of necessary prior knowledge about model discrepancy in Brynjarsdóttir and Ohagan, 2014.

The challenges mentioned related to large datasets with unknown observation uncertainty, combined with model structures simplifying the real world due to scale issues, data availability, or computational limitations, also occur in other areas than

distributed hydrological models as in this example. Hence, CRPS-based performance criteria are also expected to work in other modelling contexts and tackle the mentioned issues with squared error-based performance criteria.

**Code availability**

A python script used to calculate the CRPS as described in the article in a manner that allows communication with the PEST software package will be made publicly accessible upon publication. The script reads observed and simulated groundwater

heads from model outputs and will be accompanied by example files.

**Author contribution**

The necessity to deal with inadequate observations was recognized by all co-authors, when Raphael Schneider brought up the idea of using a CRPS-based objective function. Subsequently, the idea was developed in discussions of all co-authors. Most of the modelling work was performed by Raphael Schneider with support from Simon Stisen. Raphael Schneider and Simon

Stisen prepared the manuscript with contributions from Hans Jørgen Henriksen.

**Competing interests**

The authors declare that they have no conflict of interest.





## Acknowledgements

We want to gratefully acknowledge the "FODS 6.1 fasttrack metodeudvikling" project headed by Styrelsen for Dataforsyning
og Effektivisering (SDFE) (Danish Agency for Data Supply and Efficiency), in which context the presented method was
developed. Furthermore, we want to thank Jens Christian Refsgaard for kindly providing helpful feedback to improve the
manuscript.

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





**Table 1: Performance of the two models and three different objective functions after calibration using the different objective functions (with initial parameter set 1). For the groundwater head performance (given as obs – sim), lowest values for each metric are marked bold. For the discharge performance, mean absolute values are given across the 9 or 6 stations for Odense and Storå. Water balance error (fbal) given as (obs – sim)/obs.**

| performance metrics | objective function used | Odense | | | Storå | | |
|---|---|---|---|---|---|---|---|
| | | CRPS | MAE | MSE | CRPS | MAE | MSE |
| groundwater head | ME | **0.31** | -0.59 | -0.62 | **-0.04** | -0.65 | -0.71 |
| | MAE | 3.19 | **3.02** | 3.05 | 2.61 | 2.62 | **2.61** |
| | RMSE | 6.13 | 5.94 | **5.91** | 4.06 | 3.96 | **3.90** |
| | CRPS | **0.72** | 0.84 | 0.86 | **0.66** | 0.75 | 0.77 |
| river discharge | NSE | 0.75 | 0.76 | 0.76 | 0.63 | 0.64 | 0.65 |
| | fbal | 7.4% | 8.0% | 8.4% | 3.0% | 2.4% | 2.7% |
| | fbal_s | 14.8% | 15.6% | 17.1% | 16.4% | 14.7% | 13.8% |

**Table 2: Displaying the number of grid cells that have lower error after CRPS calibration than after MSE calibration ($ME_i$ better), compared to the number of grid cells that have higher error after CRPS calibration than after MSE calibration ($ME_i$ worse). Grid cells are grouped according to their error after CRPS calibration.**

| $|ME_i|$ in CRPS calibration | Odense | | | Storå | | |
|---|---|---|---|---|---|---|
| | $ME_i$ better | $ME_i$ worse | total | $ME_i$ better | $ME_i$ worse | total |
| 0 – 5m | 541 | 289 | 830 | 288 | 178 | 466 |
| 5 – 10m | 27 | 59 | 86 | 32 | 31 | 63 |
| > 10m | 6 | 43 | 49 | 9 | 15 | 24 |
| total | 574 | 391 | 965 | 329 | 224 | 553 |

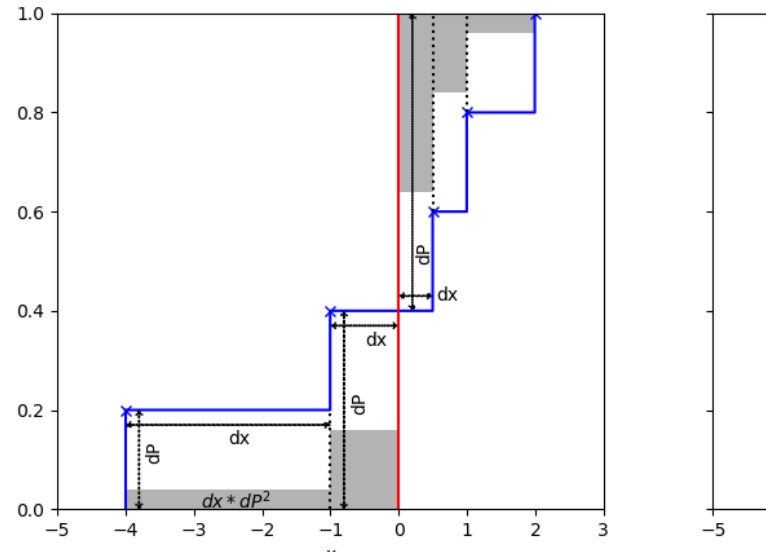 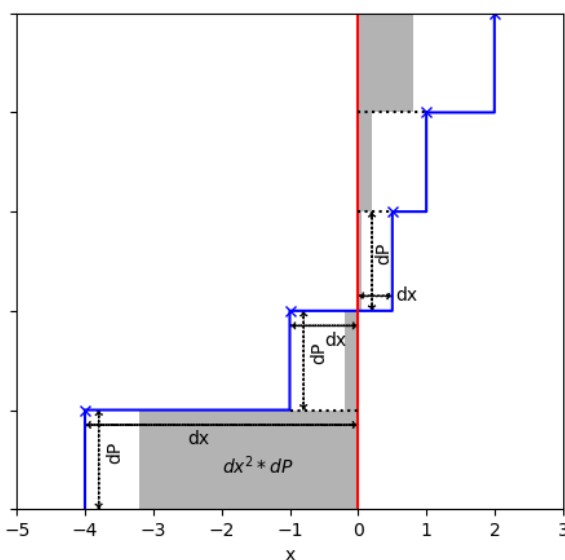

**Figure 1: Illustration of the calculation of the CRPS (left panel) and MSE (right panel) for a simple cumulative distribution of a dataset of five predictions of the variable *x* (blue line and crosses) with a true value of 0 (red line). The sum of the grey areas results in the CRPS or MSE, respectively. Note the different bases for *dx* and *dP*, which are shown by splitting the areas along the x- and y-direction, respectively.**





**Figure 2: Basemap of the two study areas Storå (A), Odense (B) and their location within Denmark (C). The model boundaries and stream network of the two MIKE SHE models are displayed, together with the discharge stations and groundwater wells with head observations used in the calibration.**



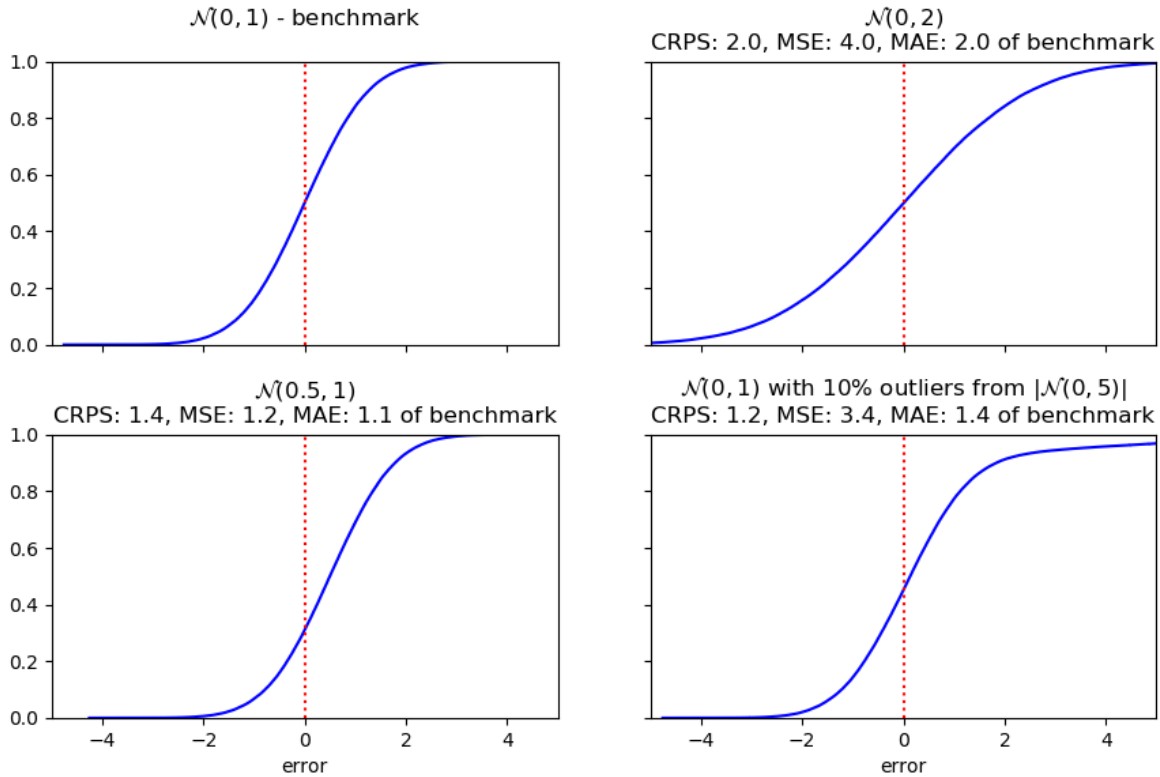

**Figure 3: Sensitivity of the CRPS in comparison to MSE and MAE towards bias and outliers. The top left plot shows the benchmark. The other plot title report values of each metric relative to their respective benchmark values.**







**Figure 4: ECDF of ME per grid cell for the Odense model (left panels) and the Storå model (right panels) after calibration to CRPS, MAE, and MSE. Compared for different initial parameter sets (whose MEs are shown in grey dashed lines).**

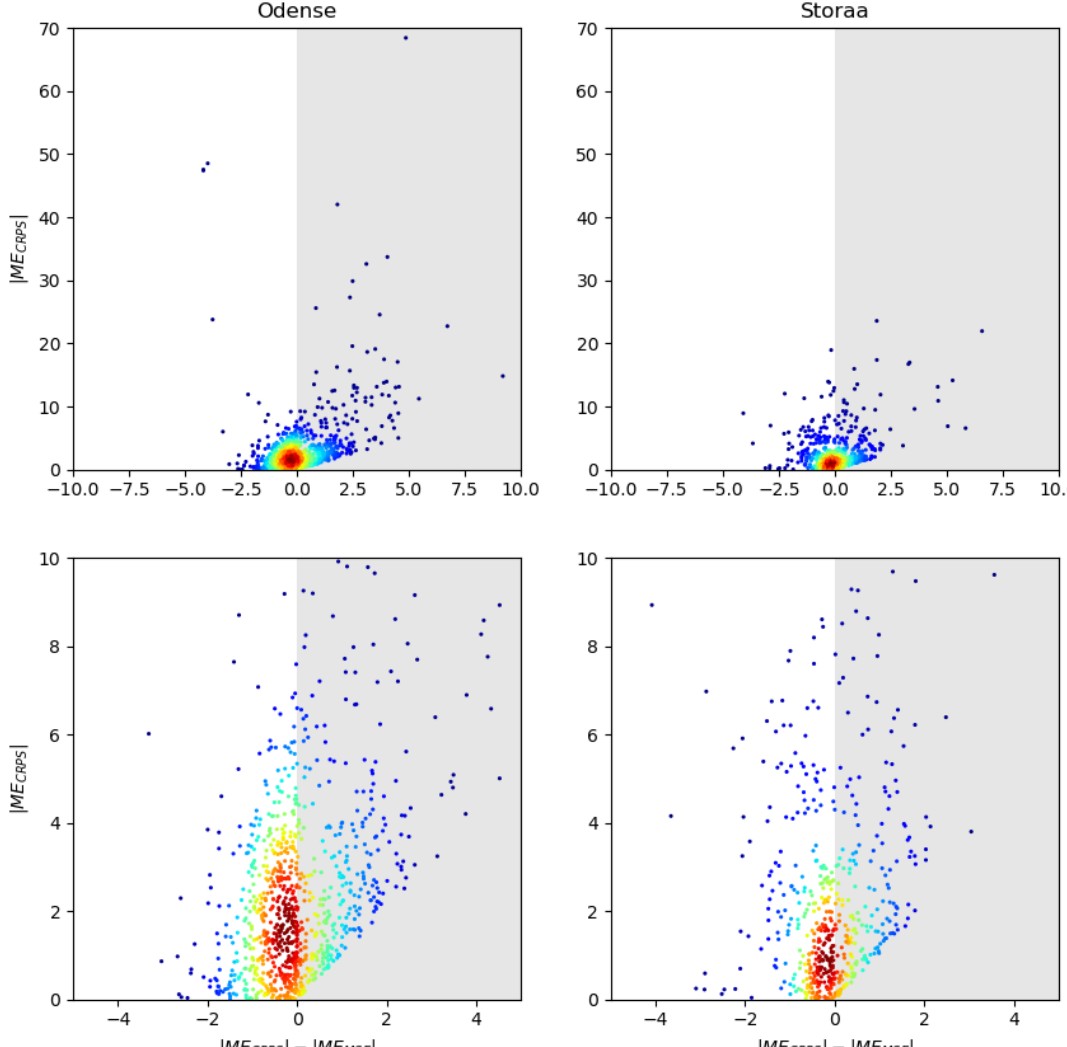

**Figure 5: Scatterplot of the difference in ME$_i$ per grid cell between the CRPS and MSE calibrations (x-axis) vs. the absolute ME per grid cell in the CRPS calibration (y-axis). The grey half of each plot contains the grid cells with a worse fit, and the white half contains the grid cells with a better fit in the CRPS calibration compared to the MSE calibration. The upper and lower panels show each show the same data; the lower panels are zoomed to a smaller region along the y-axis.**

570





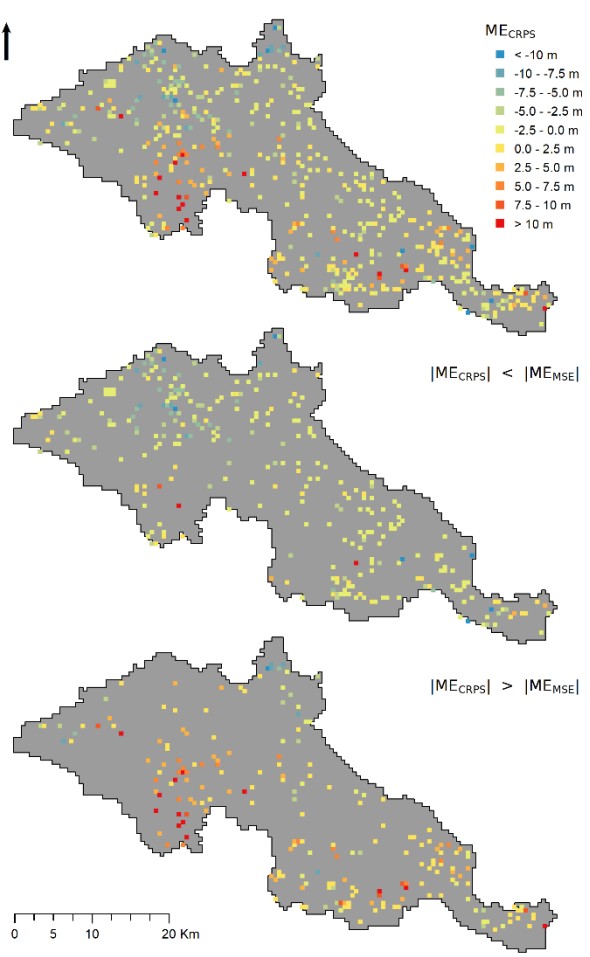

**Figure 6: ME per grid cell for the Storå model. Top: All ME per grid from the CRPS calibration. Middle: only showing ME per grid, where the CRPS performs better than the MSE calibration. Bottom: only showing ME per grid, where the CRPS calibration performs worse than the MSE calibration.**





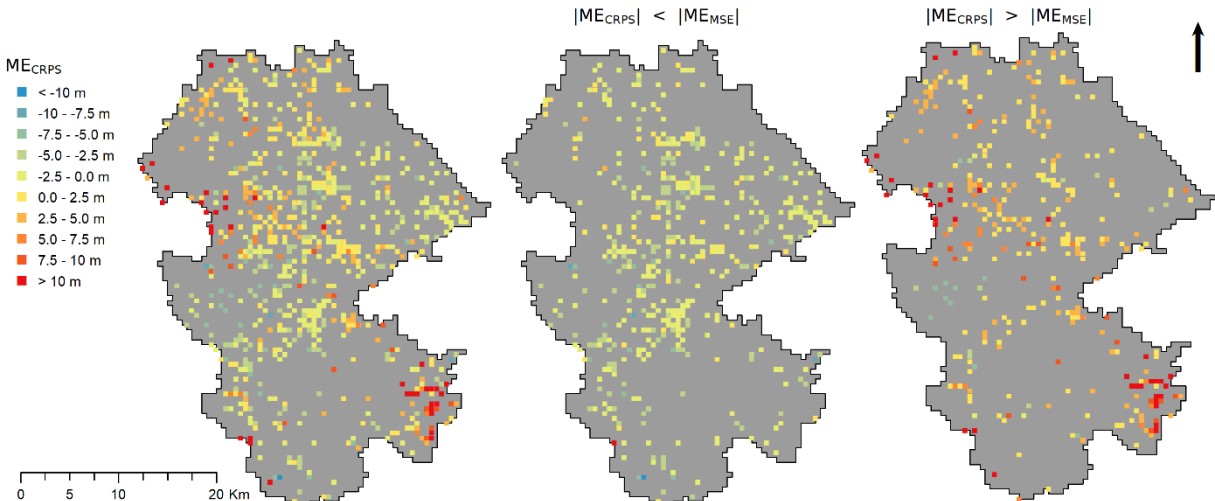

**Figure 7: ME per grid cell for the Odense model. Left: All ME per grid from the CRPS calibration. Middle: only showing ME per grid, where the CRPS performs better than the MSE calibration. Right: only showing ME per grid, where the CRPS calibration performs worse than the MSE calibration.**

580