# Peer review of "A robust objective function for calibration of groundwater models in light of deficiencies of model structure and observations"

_Hydrology and Earth System Sciences, 2019_

## Referee Comment (RC1) · Shlomo P. Neuman (Referee) · 6 Feb 2020

Comments by Shlomo P. Neuman on

A robust objective function for calibration of groundwater models in light of deficiencies of model structure and observations

by Raphael Schneider, Hans Jørgen Henriksen, and Simon Stisen

The authors propose using Continuous Ranked Probability Score (CRSP) as a criterion for calibrating groundwater models against hydraulic head data. They reason that large residual calibration errors, which often result in part from structural model errors, would dominate CRSP calibration results to a lesser degree than they do when one uses standard criteria such as mean square error (MSE) or mean absolute error (MAE); CSRP would assign greater weight to the majority of smaller residuals than to a few larger residuals at the edges of their cumulative distribution. Whereas CSRP is designed to work with ensembles of predictions, the authors suggest applying it to a single realization of calibration errors across a model space-time horizon. To test their idea, the authors apply CSRP to two regional scale coupled surface-groundwater models to conclude that their proposed criterion results in lesser calibration bias than do MSE or MAE.

I find the idea of using CRSP as a calibration criterion interesting but consider the authors' attempt to demonstrate its utility unconvincing. My reasons are as follows:

1. Applying the probabilistic CRSP criterion to a single realization of calibration error requires an assumption of ergodicity. There is no discussion of this potential restriction in the manuscript.
2. Groundwater flow models differ fundamentally from most surface water models in that parameters entering the former (hydraulic conductivity or transmissivity, specific storage or drainable porosity) tend to have reasonably well-defined physical meanings and can often be estimated, independently of the calibrated model, through methods such as pumping tests and geostatistical interpolation. This makes it possible, and often necessary, to regularize the model calibration process with the aid of parameter plausibility criteria based either on such independent prior parameter estimates or on functional criteria such as smoothness. One purpose of such regularization criteria is to ensure that large calibration errors do not dominate the parameter estimation process. Would CRSP be still necessary, and/or useful, in this context? The manuscript does not address this question.
3. The two case studies fail to provide information about the reliability of parameters estimated using either CSRP, MSE or MAE. To validly compare these three criteria, one would need to test them on synthetic systems having known structures, parameters and forcing terms that are corrupted by known random and/or systematic errors of realistic kinds and magnitudes. One would further need to explore CSRP in the context of regularization criteria such as those commonly used in groundwater model calibration. Only then would it make sense to demonstrate the utility of CSRP on partially defined field problems such as those in the two case studies described.

---

## Referee Comment (RC2) · Timothy Ginn (Referee) · 8 Feb 2020

Review comments on "A robust objective function for calibration of groundwater models in light of deficiencies of model structure and observations" by Schnieder et al., by TR Ginn.

I entirely agree with the insightful assessment provided by Prof. Neuman, and here add some further thoughts.

Already in the abstract the reader becomes worried that the central hypothesis, that structural errors or severely erroneous observations that lead to parameter (value) compensation can be ameliorated by using the new objective function (OF) norm, will not be tested. To test this hypothesis the underlying forward models require known structural errors or severely erroneous observations. Much of the discussion in the introduction, and specifically the stated Aim of the paper (line 95ff) focuses on the impact of structural errors. These are termed scale, structural or boundary condition errors by the authors but which by binary categorization – they are not observational – are in my view structural errors. This could be tested by using the CRPS norm on synthetic models with strucutural errors but this was evidently not done, in lieu of testing real field scale groundwater models. In the final statement the methodology is only "assumed" (line 351) to yield indications of structural error. I am skeptical of this because the groundwater flow equation is a diffusion equation and a structural error in one subdomain may in fact impact (downstream) heads in a relatively distant subdomain. This often happens when recharge is poorly calibrated and the simulated heads or their gradients far away, e.g., near a distant but sole outlet boundary, depart dramatically from measured values. Thus in my view the ability of the CRPS norm to address structural errors is not demonstrated.

The conceptual foundation for the method is probabilistic and as well noted by Prof. Neuman requires an ergodic argument. E.g., one instance on line 113, "…timestep." should in my opinion say "… timestep and spatial location." which creates conceptual problems in the subsequent extention to treating individual data locations in a single realization as sources for a probabilistic ensemble. Another example is (line 126 )"… and the ECDF of residuals at every single observation point" which I do not understand, unless the authors mean "… and the ECDF of the collective set of residuals in the model." It may be possible to repackage the CRPS norm as a heuristic to avoid this often insurmountable challenge. Figure 1 seems to lead to a practical (heuristic) definition of dP as a vector of normalized cumulative cardinal number (or rank) of errors, where the cardinal counting is done from the largest +/- error, and x is actually the similarly ranked differences in errors again counting from the largest +- error. If I understand how it works the CRPS norm in this example is

$$dx \cdot dP^2 = |\varepsilon_1 - \varepsilon_2| \ \left(\frac{1}{5}\right)^2 + |\varepsilon_2| \ \left(\frac{2}{5}\right)^2 + |\varepsilon_3| \ \left(\frac{3}{5}\right)^2 + |\varepsilon_4 - \varepsilon_3| \ \left(\frac{2}{5}\right)^2 |\varepsilon_5 - \varepsilon_4| \ \left(\frac{1}{5}\right)^2$$

where the first two terms are counting from the left and the last three are counting from the right because the first two are underestimates and the last three are overestimates. This clever device seems to weigh not errors but differences between errors that are adjacent in magnitude, with weight increasing with proximity of rank order to the observed value. It could be posed as a potentially promising alternative to the MSE (L2 norm) and MAE (L1 norm) and however should be raced also against a norm which magnifies the smaller errors (e.g., an L(1/2) norm).

A few lesser issues appear in the discussion of the nature of head data and of model failure modes. In lines 50-60 or thereabouts, and elsewhere, the focus is on the number of head data ("large enough set"). I believe that it is more often the distribution of the head data that is the

more salient aspect that matters to inversion. Head data are often not uniformly distributed among or representative of the whole domain and/or the various important conductive units, especially in regions of strongly varying elevations, where most wells are in the valleys. The issue of model failure (lines 70-75; also 332) is attributed to the case when untrue parameter values (resulting from a structural problem) are obtained. It should be noted that these parameter values are already effective by construction, and their untrue values even if far removed from what a local pumping test would tell are only a problem if the model is incapable of simulating past or predicting future behavior, which is often found when the water changes direction, that is, when recharge or hydraulic boundary conditions change. At line 332 the term "nonoptimal" begs this very question. If the calibration minimizes the chosen OF norm then the parameter values are indeed optimal, at least to the mathematical inverse problem.

---

## Referee Comment (RC3) · Anonymous Referee #3 · 12 Mar 2020

Comments on the paper Hess-2019-685 entitled: "A robust objective function for calibration of groundwater models in light of deficiency of model structure and observations", by R. Schneider et al.

This work intends to show that the classical objective functions (OF) in the inversion of subsurface flow, such as the sum of squared errors (SSE) between simulated and observed heads, or the sum of absolute errors (SAE), are functions mainly dominated by a few large errors. If these errors are stemming from structural model discrepancies, then the inversion procedure would compensate on model parameters to lower the OF, but with sometimes the downside of rendering awkward or unphysical solutions.

[Figure]

Therefore, it is proposed to rely upon an OF based on the continuous ranked probability score (CRPS) reputed less sensitive to large residuals, as it measures the squared distance between the cumulated probability density (cumulated statistical distribution) of model outputs and its equivalent in terms of local observations (usually, a Heaviside function). A few examples of this reduced sensitivity to high residuals are provided on the basis of very simple examples such as a series of five values, or a continuous Gaussian distribution. Then, a comparison of CRPS, SSE, and SAE is carried out for two inversion problems dealing with actual watershed systems.

It seems interesting employing the CRPS, usually devoted to the analysis of multiple equiprobable realizations of a single variable, in the framework of a single realization of a single variable but distributed over time and space. However, in my opinion, the study partly misses its target because the applications are a priori free from model structural errors; at least, these errors are not explicitly considered in the analysis of the inverse sought solutions.

I have a few concerns of various importance with the present writing, and some specific points (in a non-exhaustive inventory) that let me think that the contribution is not mature enough for rapid publication in HESS. My suggestion is that the paper needs major revisions, including new numerical investigations, and a further complete round of review.

Regarding the main concerns:

1- The notion of structural error is not well defined. In a first approach, one could consider that structural errors are all the errors that do not directly target model parameter values. This could include: errors on the geometry of the modeled system, on initial and boundary conditions, and on source-sink terms. One could also consider that structural errors are those associated with features hardly inverted in view of their direct influence on the observed state variable. In that case, one could remove initial and boundary conditions, but also source-sink terms from the structural errors,

as these characteristics of the model can be inverted in view, here, of hydraulic head measurements. Finally, in the specific case of the reported study, the model parameterization relies upon a parameterization of the zonation type, building a "block" system with uniform parameters within each block. A flawed delineation of these blocks could also be considered as a structural error. Even better, one could suggest that for the two actual tests cases reported by the authors, errors in the delineation of uniform blocks could be the main structural error generating high residuals on heads that will never be compensated by tuning the model parameters of each block. In the end, it seems important to better state what is meant by structural errors, then deliberately generate these errors in exploratory calculations before checking on the performance of a CRPS-based objective function.

2- The two actual test cases discussed in the paper are redundant, mainly because they deal with watershed systems of the same size, with the same density of streamflow routing in their surface compartment, and a very similar density of evenly spread locations monitoring the subsurface waters. Why to report on both? The authors would have been well advised to focus on a single system, and consider that an inverse solution becomes some kind of reference problem to which structural errors are added. Here, the first structural error I would give a try would be that of a flawed parameter zonation. Then, by providing us with a metric on model parameters distinguishing values inherited from the "reference" and the "flawed" problems, some proofs that CRPS outclasses SSE and SAE could be made available.

3- My understanding is that in many locations within the modeled system, the authors (for the principle of parsimony?) lump the measurements of heads at various times and in various layers of the subsurface to build an averaged information. I doubt that this information has the sensitivity of a single observation to both parameter and structural errors. Let us take for example the case of a point measurement of head located not so far from a boundary condition. This condition is flawed and prescribes a Neumann-type boundary with prescribed fluxes instead of a Dirichlet-type condition with prescribed

head. The Neumann flux is not sufficient in the wet periods to feed the system, but too high in the dry periods, thus rendering negative (positive) errors on the head at a short distance in the winter compensated by positive (negative) errors in the summer. As a result, the structural error is not seen by the data, as would render the true Dirichlet condition able to feed the system at will. This example is just for showing that averaging various measurements is probably not a good idea to reveal that structural errors exist. I must acknowledge that I have never seen in the literature inversions taking averaged errors over large periods at some locations as the basis for an OF. I guess that it is "dangerous" to proceed that way, but probably my knowledge of the literature is not sharp enough.

4- The authors employ the same cumulative distribution of residuals to build their OF, irrespective of the location where the distribution is used to measure the performance of the model. This implies that the distribution of residuals should be stationary over space (which differs from the assumption of ergodicity associated with the inference of a CRPS on the basis of a single realization, but could also go with. . .). I doubt that in the presence of structural errors, e.g., local errors on the system geometry, or its boundary conditions, the statistical distribution of residuals would be stationary. If the authors are right, the distribution should not be stationary in being skewed toward high residual values in regions under structural errors. By the way, the CRPS should give less weight to important residuals in regions where structural errors are plaguing the convergence of the inverse problem by only tuning the model parameters.

In addition to the above general comments, I have a few specific comments (a non-exhaustive list), mainly as the consequence of lack of clarity in the writing.

1- Line 111, Eq. 1. The CRPS seems to be not well defined if it is supposed to serve as an indicator concealed in an OF. With an integral from minus infinity to plus infinity and an expected value of zero (optimal residual) the CRPS will remain the same irrespective of the location where it is applied. My understanding is that for a variable X (here a residual) and an associated bound x, the CRPS should write as the integral

between minus infinity and x of (Ps(x')-Po(x'))ˆ2dx', with Ps(x') the probability for the variable X of not exceeding the value x'. In this case, and for a residual value x at a given location, CRPS(x) measures the distance between x and zero.

2- Lines 135-143, Eqs 2 and 4... If the significance of the dPi is well exemplified in Fig. 1 (with differences between the left panel (CRPS) and the right panel (MSE)), the text does not mention this difference. In a CRPS dPi is the cumulated probability of not exceeding the value xi, when dPi in a MSE is the probability of x being within an interval bounded by xi-1 – xi, or something of the kind. I would change the notation to avoid misunderstandings and be clear on that in the main text.

3- Line 169. What means "a description of the unsaturated zone" in MIKE SHE, a simplification, a 3-D resolution of the Richards equations? A short explanation should be given as a reminder. Integrated hydrological models coupling surface and subsurface flow have many options to handle the subsurface including the vadose and the saturated zones, and very often these options condition how two different models respond differently to the same forward problem.

4- Section 3 "Model and data". As told earlier, I think that presenting a single model for a single study area would be enough. In general, the overall depiction of the models in terms of hydrological context is very poor. The reader ignores what are, for example, the mean discharges of the stream at the outlet of the system, their seasonal variability, the overall variability of heads within the subsurface, what is the hydro-meteorological forcing, what are the boundary conditions, etc. Even though the main question is not to go into the detailed features of the forward problem, a few words for fixing the context would be welcome. The hydrological context could condition the applicability of the CRPS as an OF; most of inverse problems are case-study dependents.

5- Line 175 -. It is stated that the hydrogeological model (the subsurface) encloses several "layers", which I think to be the representation of a geological stratification in the subsurface, with the consequence of generating vertical heterogeneity in the hydraulic

parameters. A few lines later, (230 and followings) it is stated that only "six different geological units' hydraulic conductivities are sought, which would mean that within a unit (a "block" sub-system), the conductivity is uniform over the various geological layers. Why to distinguish these layers in the model geometry if they are similar in terms of hydraulic parameters?

6- Line 235 and followings. The so-called "benchmark" appears here as drawn out of the blue. When the reader expects that it will be discussed on the application of the CRPS, SSE, and SAE, to the actual case-studies, a "synthetic" problem is presented based on the various responses of the OFs to continuous Gaussian distributions of residuals. In addition, the "benchmark" is not well presented at all, and the reader is required to conjecture on the calculations performed in the benchmark.

7- Section 4.2. Even though, associated tables and figures report on the fact that CRPS outperforms the other OF, all the material is in fact a blind test as we ignore what are the structural errors in the models rendering high residuals. As told earlier, I would focus on a single test-case, I would consider a given inverse solution as a reference problem, and then I would add deliberate structural errors, for example on the delineation of the unit blocks, by overestimating or under estimating the aquifer thickness is some areas, by modifying the boundary conditions, by artificially generating a few zones of preferential infiltration, etc... Then by inverting these various configurations, a comparison of the performances of the various OFs could be carried out. In the present form of the study, the CRPS appears better as a matter of fact completely dependent of the overall settings of the forward problem, but applicability to other contexts is compromised, and a better response to structural errors (even though these errors probably exist in the tested forward problems) is not proven.

---

## Author Comment (AC1) · 8 Apr 2020

(in this document, **all reviewer comments are in bold**, whereas our replies are in standard font)

**Comments by Shlomo P. Neuman on**
**A robust objective function for calibration of groundwater models in light of deficiencies of model structure and observations**
**by Raphael Schneider, Hans Jørgen Henriksen, and Simon Stisen**

**The authors propose using Continuous Ranked Probability Score (CRSP) as a criterion for calibrating groundwater models against hydraulic head data. They reason that large residual calibration errors, which often result in part from structural model errors, would dominate CRSP calibration results to a lesser degree than they do when one uses standard criteria such as mean square error (MSE)or mean absolute error(MAE); CSRP would assign greater weight to the majority of smaller residuals than to a few larger residuals at the edges of their cumulative distribution .Whereas CSRP is designed to work with ensembles of predictions, the authors suggest applying it to a single realization of calibration errors across a model space-time horizon. To test their idea, the authors apply CSRP to two regional scale coupled surface-groundwater models to conclude that their proposed criterion results in lesser calibration bias than do MSE or MAE.**

Reply:
We thank Prof. Neuman for his insightful and critical review.

For better understanding of the overall changes, we want to start with the overall plan on how to revise this manuscript (parts of these changes have been inspired by the responses by the other reviewers):
1. We will add a synthetic calibration test based on the Storå model showing the ability of the CRPS compared to other metrics
2. The synthetic test will also include the mean absolute error (MAE) and mean root error (MRE) as objective functions. The CRPS, MAE, and MRE are all performing similarly well in our test case; and clearly better than the conventional MSE. Based on the higher sensitivity towards bias of the CRPS compared to the MAE and MRE, we still prefer the CRPS.
3. We will remove the Odense from most of the presentation of the "real-world" results to reduce redundancy, and only use it for a kind of "proxy-basin" validation test

All further changes we intend to make are outlined below and in the replies to the other reviewers. Overall, we are thankful for the feedback from all three reviewers and are convinced that the manuscript will benefit significantly from the revisions.

In general, we feel it is relevant to stress that our suggested approach of using an alternative objective function to the commonly used MSE originates from dealing with issues arising in the practical application of large-scale hydrological models, and less from theoretical considerations on inverse problems etc. Such practical issues, as (i) the mentioned unavoidable structural errors in large-scale models due to a lack of detailed knowledge of the geology, process simplifications, or matters of scale, (ii) observational datasets of unknown and varying quality as well as scarce and uneven distribution in time and space, and (iii) the inability to quantify and disentangle different sources of uncertainty in many real-world problems, lead to often unsound ways of dealing with data that "do not seem to fit", i.e. a somewhat arbitrary outlier filtering. We want to provide the practitioner an objective function that better allows to also ingest some "flawed" observations – when it is not possible to detect which observations actually are flawed. Moreover, such an objective function will better allow to identify areas where our model or data is flawed, as we mention in the manuscript in lines 280ff., and then subsequently perform further investigations into the cause of the discrepancies. We feel that, in general, the sensitivity of the squared error-based objectives in inversions is acknowledged (think e.g. also of more advanced techniques of handling this issue, such as iteratively reweighted regression), but too often – at least in practical applications – is ignored, partly due

to the above outlined challenges of unknown data quality and modelling requiring stark simplifications of reality. We will make this background more clear in the revised version of the manuscript.

Though, of course we agree that also applied modelling approaches need to be examined based on a sound theoretical foundation. We are convinced that the manuscript will benefit greatly by addressing these concerns, amongst others by adding synthetic examples to further showcase the claimed benefits of the CRPS-based objective function.

**I find the idea of using CRSP as a calibration criterion interesting but consider the authors' attempt to demonstrate its utility unconvincing. My reasons are as follows:**
   1. **Applying the probabilistic CRSP criterion to a single realization of calibration error requires an assumption of ergodicity. There is no discussion of this potential restriction in the manuscript.**

Reply:
We do not assume that our calibration error is ergodic – the mean residual of a certain observation (equivalent to the mean residual of a certain ensemble member in "conventional" use of the CRPS) is not equal to the mean residual of all observations. However, we do not believe that it is a necessary criterion for using the CRPS in the way we intend, purely as a value for an objective function. There is limited literature discussing ergodicity as a strict requirement of applying the CRPS. However, we found one example arguing that the CRPS can also be used in combination with non-ergodic Schlather models (Yuen, 2015, p.14).

   2. **Groundwater flow models differ fundamentally from most surface water models in that parameters entering the former (hydraulic conductivity or transmissivity, specific storage or drainable porosity) tend to have reasonably well-defined physical meanings and can often be estimated, independently of the calibrated model, through methods such as pumping tests and geostatistical interpolation. This makes it possible, and often necessary, to regularize the model calibration process with the aid of parameter plausibility criteria based either on such independent prior parameter estimates or on functional criteria such as smoothness. One purpose of such regularization criteria is to ensure that large calibration errors do not dominate the parameter estimation process. Would CRSP be still necessary, and/or useful, in this context? The manuscript does not address this question.**

Reply:
Yes, we still consider the CRPS relevant here. For example:
   a) Distributed groundwater models' parameters have physical meaning, but because of issues with model grid scale, heterogeneity, simplification of processes (e.g. of the unsaturated zone, artificial drain, etc), errors in the hydrogeological model used to outline conductivity zones, etc. parameter values usually cannot be estimated directly. Despite not using any regularization or parameter bounds, our models' parameters fall into reasonable/expectable ranges after the calibration (e.g. horizontal conductivities in sand layers range from ~$1*10^{-3}$ m/s to ~$1*10^{-4}$ m/s, and from ~$3*10^{-6}$ m/s to ~$6*10^{-8}$ m/s in clay layers)
   b) We assume Prof. Neuman is referring mainly to highly parameterized approaches (inversion of geophysical data or highly parameterized hydrological models solved e.g. by pilot points), as he mentions regularization and smoothness (that is, smoothness of parameter fields?). We are using a unit-based approach, which does not require the use of regularization or similar. Furthermore, despite not using tight parameter bounds limiting parameters to "reasonable" values, the inversion process in the vast majority of cases ends up with "reasonable" parameter values – see point a). In pilot-point approaches, with regularization of the parameter field, one still could assume that the

CRPS is beneficial, as it still reduces the impact of single (wrong) observations on the parameter field (even if only local). However, testing this is outside the scope of this manuscript.
We will add some of these clarifications to the introduction and discussion part of the revised manuscript.

3. **The two case studies fail to provide information about the reliability of parameters estimated using either CSRP, MSE or MAE. To validly compare these three criteria, one would need to test them on synthetic systems having known structures, parameters and forcing terms that are corrupted by known random and/or systematic errors of realistic kinds and magnitudes. One would further need to explore CSRP in the context of regularization criteria such as those commonly used in groundwater model calibration. Only then would it make sense to demonstrate the utility of CSRP on partially defined field problems such as those in the two case studies described.**

Reply:
Concerning the reliability of the estimated parameter values we want to refer to our reply a) to point 2:
With the given model structure and scale, all parameter values are effective parameters, and cannot be directly measured in the field or related to literature values. However, we still have expected ranges for parameter values. In general, the estimated parameter values in the two presented real-world models fall into expected/plausible ranges, though without allowing for the conclusion that the CRPS-based objective function yields narrower confidence intervals or more "reasonable" parameter values.

Concerning synthetic tests: When developing and testing the idea of using the CRPS as an objective function, we also tested it in synthetic environments. We fully understand Prof. Neuman's concern; therefore, we will add some of the results of the synthetic tests to the revised manuscript, as outlined in the following paragraphs. Similar changes were also requested by reviewer 2, Prof. Ginn, who also was interested in a comparison of the CRPS against the mean root error (MRE) and mean absolute error (MAE) – hence, those two were also included in the synthetic tests.
(Our synthetic tests, however, are based on the same type of practically applied large-scale hydrological models as presented in the manuscript. That is, our models are parameterized based on (few) geological units, where the hydrological units are distributed based on conceptual hydrogeological models. Such model setups are common in research and practical applications. Therefore, a test of regularization is beyond the scope of our manuscript. We will add it as part of limitations/outlook to the revised manuscript)
For a synthetic test, we took a certain realization (referring to a specific set of parameters) of the Storå model as the reference model. From a run of this reference model, we sampled synthetic observations at the exact same locations and times where real observations were available. White noise with 0.5m or 1.0m standard deviation was added to the synthetic observations.
Some observations are further perturbed. (those perturbations could account for larger, but undetected observation errors or a structural error in the model leading to a local inability of the model to reproduce observed groundwater heads) In this example, all observations within a bounding box, within the uppermost five layers of the model (quaternary layers), with an observed water level of at least 5.0m below the surface were perturbed by adding 4.0m to their observed value (to their synthetic truth). These observations are displayed in Figure 1.

[Figure]

*Figure 1. All groundwater head observations in the Storå catchment. The synthetic observations are sampled from a reference model run; the observations marked with a black dot are perturbed.*

Then, starting from a different parameter set, the model is calibrated using the synthetic observations including the perturbations, with either the CRPS, MSE, MAE, or MRE as an objective function. The idea is, that the model calibrated using the CRPS as an objective function should be less affected by the few outliers in the synthetic observations. The synthetic test allows the conclusion that this actually is the case. For example, as can be seen in Figure 2, the parameter values resulting from an CRPS-based calibration are closer to the parameters of the synthetic truth than the parameters resulting from a SSE-based calibration, with the MAE and MRE-based calibrations lying in-between.

[Figure]

*Figure 2. Mean absolute deviations from true parameter values (i.e. the reference model's ones), using the perturbed dataset and different objective functions. The mean values are weighted by the relative sensitivity across the six parameters.*

Another indicator for our claim of the CRPS better coping with outliers than the SSE can be seen when comparing the model results of the reference model with each of the calibration results. This is done for the difference between average simulated groundwater head across the calibration period in the reference model compared to the models calibrated using the different objective functions, averaged across all computational layers in Figure 3. It can be seen clearly, that the calibrated model using the CRPS as an objective function is much closer to the reference model than the calibrated model using the SSE as an objective function.

The models using the MRE and MAE as an objective function perform similarly well as the CRPS. However, due to the higher sensitivity of the CRPS to biases, we prefer this objective function (also compare **Error! Reference source not found.** in the reply to reviewer 2).

We will add a discussion of the potential alternatives MRE and MAE to the revised version of the manuscript.

[Figure]

*Figure 3. The deviation of the average simulated groundwater heads [m] of the models calibrated against the perturbed observations compared to the reference model as the mean across all model layers. The ME and MAE given in each title give the average deviations across all model grid cells.*

References

Sanchez-Vila, X., Guadagnini, A. and Carrera, J.: Representative hydraulic conductivities in saturated grqundwater flow, Rev. Geophys., 44(3), 1–46, doi:10.1029/2005RG000169, 2006.

Wang, Y. L., Yeh, T. C. J., Wen, J. C., Gao, X., Zhang, Z. and Huang, S. Y.: Resolution and Ergodicity Issues of River Stage Tomography With Different Excitations, Water Resour. Res., 55(6), 4974–4993, doi:10.1029/2018WR023204, 2019.

Yuen, R. A.: Topics on estimation, prediction and bounding risk for multivariate extremes, The University of Michigan. [online] Available from:

https://deepblue.lib.umich.edu/bitstream/handle/2027.42/111408/bobyuen_1.pdf, 2015.

---

## Author Comment (AC2) · 8 Apr 2020

(in this document, **all reviewer comments are in bold**, whereas our replies are in standard font)

**Review comments on "A robust objective function for calibration of groundwater models in light of deficiencies of model structure and observations" by Schnieder et al., by TR Ginn.**

**I entirely agree with the insightful assessment provided by Prof. Neuman, and here add some further thoughts.**

**Already in the abstract the reader becomes worried that the central hypothesis, that structural errors or severely erroneous observations that lead to parameter (value) compensation can be ameliorated by using the new objective function (OF) norm, will not be tested. To test this hypothesis the underlying forward models require known structural errors or severely erroneous observations. Much of the discussion in the introduction, and specifically the stated Aim of the paper (line 95ff) focuses on the impact of structural errors. These are termed scale, structural or boundary condition errors by the authors but which by binary categorization – they are not observational – are in my view structural errors. This could be tested by using the CRPS norm on synthetic models with structural errors but this was evidently not done, in lieu of testing real field scale groundwater models. In the final statement the methodology is only "assumed" (line 351) to yield indications of structural error. I am skeptical of this because the groundwater flow equation is a diffusion equation and a structural error in one subdomain may in fact impact (downstream) heads in a relatively distant subdomain. This often happens when recharge is poorly calibrated and the simulated heads or their gradients far away, e.g., near a distant but sole outlet boundary, depart dramatically from measured values. Thus in my view the ability of the CRPS norm to address structural errors is not demonstrated.**

Reply:
We want to thank Prof. Ginn for his comments on our manuscript. We understand many of his concerns, and will address them for the revised version of the manuscript.

For better understanding of the overall changes, we want to start with the overall plan on how to revise this manuscript (parts of these changes have been inspired by the responses by the other reviewers):
1. We will add a synthetic calibration test based on the Storå model showing the ability of the CRPS compared to other metrics
2. The synthetic test will also include the mean absolute error (MAE) and mean root error (MRE) as objective functions. The CRPS, MAE, and MRE are all performing similarly well in our test case; and clearly better than the conventional MSE. Based on the higher sensitivity towards bias of the CRPS compared to the MAE and MRE, we still prefer the CRPS.
3. We will remove the Odense from most of the presentation of the "real-world" results to reduce redundancy, and only use it for a kind of "proxy-basin" validation test

All further changes we intend to make are outlined below and in the replies to the other reviewers. Overall, we are thankful for the feedback from all three reviewers and are convinced that the manuscript will benefit significantly from the revisions.

In general, we feel that it is relevant to point out (as we also did in our reply to Prof. Neuman) that our suggested use of the CRPS as an objective function originates from issues seen in practical applications (and less from theoretical considerations). In practical, real-world large-scale hydrological modelling, researchers commonly have to employ pragmatic solutions to tackle issues arising from the mentioned potential structural issues and observational errors. This even sometimes leads to the somewhat arbitrary exclusion of data because they in some way "do not seem to fit" to the model (or the conceptual understanding) – see lines 77ff. in the manuscript. This is where we hope that the CRPS (or the use of other, similar objective functions) can contribute to, for example, avoid arbitrary exclusion of data and, in general, provide us with

a more robust objective function (less prone to parameter compensation) in cases where we can almost be certain that parts of the model structure/data are flawed without being able to fully detect the specific issues. We will make this background more clear in the revised version of the manuscript.

Moving to Prof. Ginn's comments – we do agree that synthetic experiments are relevant to better show the claimed benefits of using the CRPS as an objective function. We already have performed some such experiments (without including them in the first submission), and are confident that we can show its benefits:

For a synthetic test, we took a certain realization (referring to a specific set of parameters) of the Storå model as a reference model. From a run of this reference model, we sampled synthetic observations at the exact same locations and times where real observations were available. White noise with 0.5m or 1.0m standard deviation was added to the synthetic observations.
Some observations are further perturbed. (those perturbations could account for larger, but undetected observation errors or a structural error in the model leading to a local inability of the model to reproduce observed groundwater heads) In this example, all observations within a bounding box, within the uppermost five layers of the model (quaternary layers), with an observed water level of at least 5.0m below the surface were perturbed by adding 4.0m to their observed value (to their synthetic truth). These observations are displayed in Figure 1.

[Figure]

*Figure 1. All groundwater head observations in the Storå catchment. The synthetic observations are sampled from a reference model run; the observations marked with a black dot are perturbed.*

Then, starting from a different parameter set, the model is calibrated using the synthetic observations including the perturbations, with either the CRPS, MSE, MAE, or MRE as an objective function. The idea is, that the model calibrated using the CRPS as an objective function should be less affected by the few outliers in the synthetic observations. The synthetic test allows the conclusion that this actually is the case. For example, as can be seen in Figure 2, the parameter values resulting from an CRPS-based calibration are closer to the parameters of the synthetic truth than the parameters resulting from a SSE-based calibration, with the MAE and MRE-based calibrations lying in-between.

[Figure]

*Figure 2. Mean absolute deviations from true parameter values (i.e. the reference model's ones), using the perturbed dataset and different objective functions. The mean values are weighted by the relative sensitivity across the six parameters.*

Another indicator for our claim of the CRPS better coping with outliers than the SSE can be seen when comparing the model results of the reference model with each of the calibration results. This is done for the difference between average simulated groundwater head across the calibration period in the reference model compared to the models calibrated using the different objective functions, averaged across all computational layers in Figure 3. It can be seen clearly, that the calibrated model using the CRPS as an objective function is much closer to the reference model than the calibrated model using the SSE as an objective function.

The models using the MRE and MAE as an objective function perform similarly well as the CRPS. However, due to the higher sensitivity of the CRPS to biases, we prefer this objective function (also compare Figure 4 below).

We will add a discussion of the potential alternatives MRE and MAE to the revised version of the manuscript.

[Figure]

*Figure 3. The deviation of the average simulated groundwater heads [m] of the models calibrated against the perturbed observations compared to the reference model as the mean across all model layers. The ME and MAE given in each title give the average deviations across all model grid cells.*

It is true that, as Prof. Ginn points out, structural errors in a groundwater model in some cases only can be seen at detached/downstream locations. A complete detachment of the location of the structural error in a model (i.e. false representation of geology) and the respective response in simulated values (e.g. simulated groundwater heads showing bias), however, only occurs in some specific cases (e.g. where lateral flow is the most important flow path). In those cases, it will always be hard to point out *where* exactly there is an issue with model structure, but still there is an indication that there is an issue with model structure *somewhere*. In many cases, the effects of faults in model structure will be less detached from their impacts. We will discuss such limitations in the revised version of the manuscript.

**The conceptual foundation for the method is probabilistic and as well noted by Prof. Neuman requires an ergodic argument. E.g., one instance on line 113, "…timestep." should in my opinion say "… timestep and spatial location." which creates conceptual problems in the subsequent extention to treating individual data locations in a single realization as sources for a probabilistic ensemble. Another example is (line 126 )"… and the ECDF of residuals at every single observation point" which I do not understand, unless the authors mean "… and the ECDF of the collective set of residuals in the model." It may be possible to repackage the CRPS norm as a heuristic to avoid this often insurmountable challenge. Figure 1 seems to lead to a practical (heuristic) definition of dP as a vector of normalized cumulative cardinal number (or rank) of errors, where the cardinal counting is done from the largest +/- error, and x is actually the similarly ranked differences in errors again counting from the largest +- error. If I understand how it works the CRPS norm in this example is**

$$dx * dP^2 = |\varepsilon_1 - \varepsilon_2| \left(\frac{1}{5}\right)^2 + |\varepsilon_2| \left(\frac{2}{5}\right)^2 + |\varepsilon_3| \left(\frac{3}{5}\right)^2 + |\varepsilon_4 - \varepsilon_3| \left(\frac{2}{5}\right)^2 |\varepsilon_5 - \varepsilon_4| \left(\frac{1}{5}\right)^2$$

**where the first two terms are counting from the left and the last three are counting from the right because the first two are underestimates and the last three are overestimates. This clever device seems to weigh not errors but differences between errors that are adjacent in magnitude, with weight increasing with proximity of rank order to the observed value. It could be posed as a potentially promising alternative to the MSE (L2 norm) and MAE (L1 norm) and however should be raced also against a norm which magnifies the smaller errors (e.g., an L(1/2) norm).**

Reply:
We do not assume that our calibration error is ergodic – the mean residual of a certain observation (equivalent to the mean residual of a certain ensemble member in "conventional" use of the CRPS) is not equal to the mean residual of all observations. However, we do not believe that it is a necessary criterion for using the CRPS in the way we intend, purely as a value for an objective function. There is limited literature discussing ergodicity as a strict requirement of applying the CRPS. However, we found one example arguing that the CRPS can also be used in combination with non-ergodic Schlather models (Yuen, 2015, p.14).

Yes, by our statement in line 126 we mean the ECDF of the collective set of residuals in the model. We will clarify this in the revised manuscript.

We agree with the reviewer's understanding of the CRPS, as shown for the example case with five values in Figure 1 and are glad to hear that he as well considers it a potentially promising alternative to the commonly used MSE or MAE.

We find the reviewer's idea with the L(1/2) norm (mean root error, MRE) interesting, and will include it in the synthetic example, see above. We still want to focus on the CRPS in this manuscript, given its higher sensitivity to a general bias, as can be seen in Figure 4 below. The RME and CRPS may be similarly sensitive to outliers (see the bottom right plot). However, as can be seen in the bottom left plot, the MRE is less sensitive to a systematic bias than the CRPS. In our models, where we are keen to achieve general water balance error, we Figure 4. Sensitivity of the CRPS in comparison to MSE, MAE, and RME towards bias and outliers. The top left plot shows the reference. The other plot titles report values of each metric relative to their respective reference values.prefer the behavior of the CRPS. A discussion of this will be added to the revised manuscript alongside Figure 4, which will be replacing Figure 3 in the manuscript.

[Figure]

*Figure 4. Sensitivity of the CRPS in comparison to MSE, MAE, and RME towards bias and outliers. The top left plot shows the reference. The other plot titles report values of each metric relative to their respective reference values.*

**A few lesser issues appear in the discussion of the nature of head data and of model failure modes. In lines 50-60 or thereabouts, and elsewhere, the focus is on the number of head data ("large enough set"). I believe that it is more often the distribution of the head data that is the more salient aspect that matters to inversion. Head data are often not uniformly distributed among or representative of the whole domain and/or the various important conductive units, especially in regions of strongly varying elevations, where most wells are in the valleys. The issue of model failure (lines 70-75; also 332) is attributed to the case when untrue parameter values (resulting from a structural problem) are obtained. It should be noted that these parameter values are already effective by construction, and their untrue values even if far removed from what a local pumping test would tell are only a problem if the model is incapable of simulating past or predicting future behavior, which is often found when the water changes direction, that is, when recharge or hydraulic boundary conditions change. At line 332 the term "nonoptimal" begs this very question. If the calibration minimizes the chosen OF norm then the parameter values are indeed optimal, at least to the mathematical inverse problem.**

Reply:
Yes, we agree with the reviewer that often, the available head measurements are not distributed evenly across the model domain or the different geologic units. In Danish landscapes, however, with their rather gentle topography, at least there is no large imbalance in the distribution of head observations between valleys and ridges.
It is correctly noted, that we use the term "non-optimal" a bit loosely – we will clarify this in the revised version of the manuscript, making clear that what we want to avoid is parameter compensation.

Furthermore, we want to add that parameter compensation is not limited to model structural issues, but can also occur to compensate for observational errors.

References:

Yuen, R. A.: Topics on estimation, prediction and bounding risk for multivariate extremes, The University of Michigan. [online] Available from:
https://deepblue.lib.umich.edu/bitstream/handle/2027.42/111408/bobyuen_1.pdf, 2015.

---

## Author Comment (AC3) · 8 Apr 2020

(in this document, **all reviewer comments are in bold**, whereas our replies are in standard font)

**Comments on the paper Hess-2019-685 entitled: "A robust objective function for calibration of groundwater models in light of deficiency of model structure and observations", by R. Schneider et al.**

Reply:
We would like to thank the reviewer for their thorough review.

For better understanding of the overall changes, we want to start with the overall plan on how to revise this manuscript (parts of these changes have been inspired by the responses by the other reviewers):

1. We will add a synthetic calibration test based on the Storå model showing the ability of the CRPS compared to other metrics
2. The synthetic test will also include the mean absolute error (MAE) and mean root error (MRE) as objective functions. The CRPS, MAE, and MRE are all performing similarly well in our test case; and clearly better than the conventional MSE. Based on the higher sensitivity towards bias of the CRPS compared to the MAE and MRE, we still prefer the CRPS.
3. We will remove the Odense from most of the presentation of the "real-world" results to reduce redundancy, and only use it for a kind of "proxy-basin" validation test

All further changes we intend to make are outlined below and in the replies to the other reviewers. Overall, we are thankful for the feedback from all three reviewers and are convinced that the manuscript will benefit significantly from the revisions.

We believe that we can add significantly to the manuscript by incorporating some changes suggested by the reviewer, and improve where the reviewer pointed out some lack of clarity. Our specific replies follow below.

**This work intends to show that the classical objective functions (OF) in the inversion of subsurface flow, such as the sum of squared errors (SSE) between simulated and observed heads, or the sum of absolute errors (SAE), are functions mainly dominated by a few large errors. If these errors are stemming from structural model discrepancies, then the inversion procedure would compensate on model parameters to lower the OF, but with sometimes the downside of rendering awkward or unphysical solutions. Therefore, it is proposed to rely upon an OF based on the continuous ranked probability score (CRPS) reputed less sensitive to large residuals, as it measures the squared distance between the cumulated probability density (cumulated statistical distribution) of model outputs and its equivalent in terms of local observations (usually, a Heaviside function). A few examples of this reduced sensitivity to high residuals are provided on the basis of very simple examples such as a series of five values, or a continuous Gaussian distribution. Then, a comparison of CRPS, SSE, and SAE is carried out for two inversion problems dealing with actual watershed systems.**
**It seems interesting employing the CRPS, usually devoted to the analysis of multiple equiprobable realizations of a single variable, in the framework of a single realization of a single variable but distributed over time and space. However, in my opinion, the study partly misses its target because the applications are a priori free from model structural errors; at least, these errors are not explicitly considered in the analysis of the inverse sought solutions.**

Reply:
We would like to point out again that our suggested use of the CRPS as an objective function instead of the SSE due to its lower sensitivity to large residuals is not only owing possible structural errors, but also uncertainties in the observational data used as target in the inversion. As we for example write in lines 55 ff, we are confronted with data of unknown quality. Almost inevitably, there will be some (significant) observation errors in some of the observations, however we often cannot identify them. Hence, we argue

that there is a point in having an objective function that is less sensitive to large residuals. One alternative, that has been used sometimes (we provide some examples from literature in line 79), is to discard observations that deviate too much from the modelled values or a conceptual understanding of the modelled system. We think that this is an arbitrary, and, hence, undesired way of filtering data, potentially removing valuable information from the inversion or the later model validation. Instead, we want to use an objective function less sensitive to large outliers.

In our revised manuscript we will be more clear about the two issues we are tackling: i) structural errors, and ii) observational errors.

It is correct that the structural errors are not explicitly considered in our models; this is not/hardly possible for such large-scale practical applications. However, structural errors will still have an impact on the achievable model fit in parts of the model (for example due to a missing geological layer/lens, or a wrong boundary condition). Again, here we hope that an objective function less sensitive to large residuals will yield in a better solution of the parameter estimation.

**I have a few concerns of various importance with the present writing, and some specific points (in a non-exhaustive inventory) that let me think that the contribution is not mature enough for rapid publication in HESS. My suggestion is that the paper needs major revisions, including new numerical investigations, and a further complete round of review.**
**Regarding the main concerns:**
  1- **The notion of structural error is not well defined. In a first approach, one could consider that structural errors are all the errors that do not directly target model parameter values. This could include: errors on the geometry of the modeled system, on initial and boundary conditions, and on source-sink terms. One could also consider that structural errors are those associated with features hardly inverted in view of their direct influence on the observed state variable. In that case, one could remove initial and boundary conditions, but also source-sink terms from the structural errors, as these characteristics of the model can be inverted in view, here, of hydraulic head measurements. Finally, in the specific case of the reported study, the model parameterization relies upon a parameterization of the zonation type, building a "block" system with uniform parameters within each block. A flawed delineation of these blocks could also be considered as a structural error. Even better, one could suggest that for the two actual tests cases reported by the authors, errors in the delineation of uniform blocks could be the main structural error generating high residuals on heads that will never be compensated by tuning the model parameters of each block. In the end, it seems important to better state what is meant by structural errors, then deliberately generate these errors in exploratory calculations before checking on the performance of a CRPS-based objective function.**

Reply:
We understand that there is a need to be more clear around our use of the term "structural error" – we will clarify this in the revised manuscript. Based on our experience, though, the delineation of the geological zonation together with drainage representation and unsaturated zone description should be the largest contributor to structural uncertainty. The influence of initial conditions should be negligible, as we are considering hotstart and warmup periods when running our models. Boundary conditions have an impact, however we expect it to be moderate: along land-boundaries of the model they are taken from larger models; the sea boundary conditions are straight forward (fixed head = 0m).

In general, we consider all types of structural errors the reviewer mentioned – however, most of those cannot be properly disentangled in large-scale hydrologic models in practical applications such as ours.

2- **The two actual test cases discussed in the paper are redundant, mainly because they deal with watershed systems of the same size, with the same density of streamflow routing in their surface compartment, and a very similar density of evenly spread locations monitoring the subsurface waters. Why to report on both? The authors would have been well advised to focus on a single system, and consider that an inverse solution becomes some kind of reference problem to which structural errors are added. Here, the first structural error I would give a try would be that of a flawed parameter zonation. Then, by providing us with a metric on model parameters distinguishing values inherited from the "reference" and the "flawed" problems, some proofs that CRPS outclasses SSE and SAE could be made available.**

Reply:
We do understand the reviewer's concern regarding the redundancy of the two case studies. The main reasons for including both cases is that we (i) wanted to show some level of reproducibility and robustness, and (ii) the two cases, geologically speaking, actually are different, with the Storå catchment being relatively flat and more dominated by sand in the uppermost layers, and the Odense catchment being more hilly and dominated by clay. Also in light of extending the paper by adding a synthetic experiments (based on the Storå model), we plan to follow the reviewer's suggestion and remove most of the results of the Odense model in the revised version of the manuscript. I.e. remove the Odense results and respective discussion from Table 1 and 2 and Figure 4 and 5. We plan, however, to keep Figure 7, though potentially simplify or merge it with Figure 6. The motivation will be mentioned more explicitly in the revised manuscript.

Furthermore, we will add a synthetic experiment, based on the Storå model case, to show that the CRPS performs as stated in cases with flawed observations. These experiments are performed with synthetic observations sampled at the exact same locations and points in time as the real observations. The synthetic observations are sampled from a model run with a specific parameter set, and some white noise is added to all of them to account for general measurement uncertainty. Moreover, some of the observations are perturbed further with a systematic bias. Then, the model is calibrated against the perturbed synthetic observations, starting from a different parameter set. The calibration experiments are carried out with either the CRPS, mean root error (MRE), mean absolute erroer (MAE), or SSE as an objective function. In these synthetic calibration experiments, it could be shown that the CRPS behaves as expected by us – the parameter set resulting from the CRPS-based calibration is closer to the parameter set of the synthetic truth, than the parameter set resulting from the SSE-based calibration. For more details, we would like to refer to the reply to reviewers 1 and 2 who also mentioned the need for synthetic experiments.

3- **My understanding is that in many locations within the modeled system, the authors (for the principle of parsimony?) lump the measurements of heads at various times and in various layers of the subsurface to build an averaged information. I doubt that this information has the sensitivity of a single observation to both parameter and structural errors. Let us take for example the case of a point measurement of head located not so far from a boundary condition. This condition is flawed and prescribes a Neumann-type boundary with prescribed fluxes instead of a Dirichlet-type condition with prescribed head. The Neumann flux is not sufficient in the wet periods to feed the system, but too high in the dry periods, thus rendering negative (positive) errors on the head at a short distance in the winter compensated by positive (negative) errors in the summer. As a result, the structural error is not seen by the data, as would render the true Dirichlet condition able to feed the system at will. This example is just for showing that averaging various measurements is probably not a good idea to reveal that structural errors exist. I must acknowledge that I have never seen in the literature inversions taking averaged errors over large**

**periods at some locations as the basis for an OF. I guess that it is "dangerous" to proceed that way, but probably my knowledge of the literature is not sharp enough.**

Reply:

We do lump the head measurements to model grid cells. This means that we lump observations from various wells, if they fall into the same model grid cell, and we lump multiple observations in time per well. However, we do not lump observations across various layers of the subsurface; observations from different model layers will fall into different model grid cells, and hence will remain individual observations. This will be made more clear in the revised manuscript to avoid misunderstandings.

We do agree with the reviewer's concern that aggregating the information within each grid cell from different points in time risks the loss of some information, in particular the potential for a compensation of negative and positive residuals. However, this can only occur in very specific cases, where (i) a timeseries is available (ii) without any bias in the simulated groundwater heads, but only an error in the simulated groundwater head amplitudes (usually from seasonal effects).

However, we try to explain our motivation for such an aggregation:

- We do only aggregate to the smallest spatial unit the model can resolve – a single model cell.
- In many cases, we aggregate single observations (not time series) from different wells within one model grid. These observations actually often "contradict" each other, for example by showing a positive residual of a few metres in the one well, and a negative residual of a few metres in the other. Reasons for that can be anything from heterogeneities within one model cell not being described in the model, small scale topographic variability, observations errors, etc. We cannot identify the exact reason, or identify which observations are most valid. Therefore, we still want to use all available data, and assume that it is reasonable to trust the aggregated mean per model grid cell in this case.
- Typical seasonal variations are in the range of ~1m (or less), whereas our residuals are in the range of a few metres. I.e. our typical model residuals are significantly larger than seasonal variations, which reduces the information lost from aggregation in time.

In practical applications, at least with large-scale models and large datasets of varying origin, quality, and spatiotemporal resolution it seems common to, for example, aggregate all observations from one well into one residual, i.e. aggregate over time (Sonnenborg et al., 2003).

4- **The authors employ the same cumulative distribution of residuals to build their OF, irrespective of the location where the distribution is used to measure the performance of the model. This implies that the distribution of residuals should be stationary over space (which differs from the assumption of ergodicity associated with the inference of a CRPS on the basis of a single realization, but could also go with…). I doubt that in the presence of structural errors, e.g., local errors on the system geometry, or its boundary conditions, the statistical distribution of residuals would be stationary. If the authors are right, the distribution should not be stationary in being skewed toward high residual values in regions under structural errors. By the way, the CRPS should give less weight to important residuals in regions where structural errors are plaguing the convergence of the inverse problem by only tuning the model parameters.**

Reply:

Unfortunately, we are unsure whether we fully understand this comment. We will try to respond to the reviewer, but also ask them for clarification if we misunderstood.

The CRPS is calculated across all residuals in the model. It is correct that we do not assume that the residuals are distributed stationary over space (concerning ergodicity, please refer to the answer to the first comment of reviewer 1).

Concerning the last sentence in this comment: Yes, that is exactly our point: The CRPS should give less weight to large residuals in regions with structural errors (or observation errors). Hence, it is less prone to parameter compensation, potentially allowing an easier identification of such areas in the model after the calibration.

**In addition to the above general comments, I have a few specific comments (a nonexhaustive list), mainly as the consequence of lack of clarity in the writing.**

1- **Line 111, Eq. 1. The CRPS seems to be not well defined if it is supposed to serve as an indicator concealed in an OF. With an integral from minus infinity to plus infinity and an expected value of zero (optimal residual) the CRPS will remain the same irrespective of the location where it is applied. My understanding is that for a variable X (here a residual) and an associated bound x, the CRPS should write as the integral between minus infinity and x of (Ps(x')-Po(x'))ˆ2dx', with Ps(x') the probability for the variable X of not exceeding the value x'. In this case, and for a residual value x at a given location, CRPS(x) measures the distance between x and zero.**

Reply:
Equation 1 gives the general, original definition of the CRPS (except for formalities identical to the formulation in the cited (Gneiting and Raftery, 2005) and (Hersbach, 2000)). We are not sure we do fully comprehend the reviewer's concern. Maybe there is some confusion arising from the fact that, usually, the CRPS is applied to the difference between Ps and Po, where Po is the true/observed value – whereas in our case, as we are dealing with residuals and not absolute values, Po is zero.
We think the reviewer could have misunderstood how we apply the CRPS (also concerning the comment above) – it is not applied to every "location", but is applied across all observations of the entire model. We are happy and confident to fully clear this up with the reviewer in the following round.

2- **Lines 135-143, Eqs 2 and 4. If the significance of the dPi is well exemplified in Fig. 1 (with differences between the left panel (CRPS) and the right panel (MSE)), the text does not mention this difference. In a CRPS dPi is the cumulated probability of not exceeding the value xi, when dPi in a MSE is the probability of x being within an interval bounded by xi-1 – xi, or something of the kind. I would change the notation to avoid misunderstandings and be clear on that in the main text.**

Reply:
Yes, that is correctly understood. This is mentioned in the figure caption, though maybe not clear enough. Will be explained better in the text as well.

3- **Line 169. What means "a description of the unsaturated zone" in MIKE SHE, a simplification, a 3-D resolution of the Richards equations? A short explanation should be given as a reminder. Integrated hydrological models coupling surface and subsurface flow have many options to handle the subsurface including the vadose and the saturated zones, and very often these options condition how two different models respond differently to the same forward problem.**

Reply:
Yes, the description of the unsaturated zone is a crucial part of such coupled models, we will add a few more details in the revised version of the manuscript. The described models are using the 2-Layer method of MIKE SHE. This is a relatively simple description of the unsaturated zone including the processes of interception, ponding and evapotranspiration while simplifying the entire unsaturated zone to two layers (DHI, 2019, p.27).

4- **Section 3 "Model and data". As told earlier, I think that presenting a single model for a single study area would be enough. In general, the overall depiction of the models in terms of hydrological context is very poor. The reader ignores what are, for example, the mean discharges of the stream at the outlet of the system, their seasonal variability, the overall variability of heads within the subsurface, what is the hydro-meteorological forcing, what are the boundary conditions, etc. Even though the main question is not to go into the detailed features of the forward problem, a few words for fixing the context would be welcome. The hydrological context could condition the applicability of the CRPS as an OF; most of inverse problems are case-study dependents.**

Reply:
We left out further descriptions of the model areas because we considered those things covered by the various publications on our national model, which the models presented in this manuscript are based on. But we certainly can understand, that further details will give a better context to most readers who are unfamiliar with Denmark, without having to refer to the references. We will add further information along the lines of the reviewer's suggestions in the revised manuscript.

5- **Line 175 -. It is stated that the hydrogeological model (the subsurface) encloses several "layers", which I think to be the representation of a geological stratification in the subsurface, with the consequence of generating vertical heterogeneity in the hydraulic parameters. A few lines later, (230 and followings) it is stated that only "six different geological units' hydraulic conductivities are sought, which would mean that within a unit (a "block" sub-system), the conductivity is uniform over the various geological layers. Why to distinguish these layers in the model geometry if they are similar in terms of hydraulic parameters?**

Reply:
Given the aggregation of some of the layers of the hydrogeological model into fewer (seven) computational layers in the model, there not only is a vertical heterogeneity, but also a horizontal heterogeneity within each computational layer.
That means, that despite only calibrating six different geological units, there is heterogeneity in across the seven computational layers. Furthermore, some of the geological units span several distinct layers, for example can we have a case where layer 1 (top) is a sand layer with conductivity K1, overlaying a clay layer (middle) with conductivity K2, overlaying itself a sand layer (bottom) with conductivity K1 again.

6- **Line 235 and followings. The so-called "benchmark" appears here as drawn out of the blue. When the reader expects that it will be discussed on the application of the CRPS, SSE, and SAE, to the actual case-studies, a "synthetic" problem is presented based on the various responses of the OFs to continuous Gaussian distributions of residuals. In addition, the "benchmark" is not well presented at all, and the reader is required to conjecture on the calculations performed in the benchmark.**

Reply:
The mentioned section (lines 235 ff. and Figure 3) is meant as an illustration of the general behavior of the different objective functions (CRPS, SSE, SAE) to different example distributions of residuals. The "benchmark" mentioned is just a normal distribution with mean 0 and standard deviation 1, and its meant as a reference to be compared to the other distributions, which are slightly deviating from it (either by introducing a bias or adding some outliers. Maybe the term "reference" is better than "benchmark"?

Furthermore, we understand that section 4.1 is not as such a result of our hydrological model calibration/the application of the CRPS as an objective function to our hydrological model. Hence, section 4.1 and the related Figure 3 could be moved to the end of section 2: After explaining the CRPS in general, its behavior to large values and its differences to the SSE, those effects can be shown based on a few example distributions (Figure 3).

**7- Section 4.2. Even though, associated tables and figures report on the fact that CRPS outperforms the other OF, all the material is in fact a blind test as we ignore what are the structural errors in the models rendering high residuals. As told earlier, I would focus on a single test-case, I would consider a given inverse solution as a reference problem, and then I would add deliberate structural errors, for example on the delineation of the unit blocks, by overestimating or under estimating the aquifer thickness is some areas, by modifying the boundary conditions, by artificially generating a few zones of preferential infiltration, etc… Then by inverting these various configurations, a comparison of the performances of the various OFs could be carried out. In the present form of the study, the CRPS appears better as a matter of fact completely dependent of the overall settings of the forward problem, but applicability to other contexts is compromised, and a better response to structural errors (even though these errors probably exist in the tested forward problems) is not proven.**

Reply:
These are all valid concerns. We added a synthetic test (see reply to point 2). Furthermore, we want to point out again (as in our reply to the initial comment of the reviewer) that we are not only considering potential errors arising from structural errors, but also observational errors.

References

DHI: MIKE SHE, Volume 2: Reference Guide, , 2, 374 [online] Available from: https://manuals.mikepoweredbydhi.help/2019/Water_Resources/MIKE_SHE_Printed_V2.pdf, 2019.

Gneiting, T. and Raftery, A. E.: Strictly Proper Scoring Rules, Predictions, and Estimation., 2005.

Hersbach, H.: Decomposition of the Continuous Ranked Probability Score for Ensemble Prediction Systems, Weather Forecast., 15(5), 559–570, doi:10.1175/1520-0434(2000)015<0559:DOTCRP>2.0.CO;2, 2000.

Sonnenborg, T. O., Christensen, B. S. B., Nyegaard, P., Henriksen, H. J. and Refsgaard, J. C.: Transient modeling of regional groundwater flow using parameter estimates from steady-state automatic calibration, J. Hydrol., 273(1–4), 188–204, doi:10.1016/S0022-1694(02)00389-X, 2003.